# SWE-Dev: Evaluating and Training Autonomous End-to-End Feature-Driven Software Development

## Abstract

Large Language Models (LLMs) have shown strong capability in diverse software engineering tasks, e.g. code completion, bug fixing, and document generation. However, feature-driven development, a highly prevalent real-world task that involves developing new functionalities for large, existing codebases, remains underexplored. We therefore introduce SWE-Dev, the first large-scale dataset (with 14,000 training and 500 test samples) designed to evaluate and train autonomous coding systems on real-world end-to-end feature-driven software development tasks. To ensure verifiable and diverse training, SWE-Dev uniquely provides all instances with a runnable environment and its developer-authored executable unit tests. This collection not only provides high-quality data for Supervised Fine-Tuning (SFT), but also enables Reinforcement Learning (RL) by delivering accurate reward signals from executable unit tests. Our extensive evaluations on SWE-Dev, covering 17 chatbot LLMs, 10 reasoning models, and 10 Multi-Agent Systems (MAS), reveal that feature-driven software development is a profoundly challenging frontier for current AI (e.g., Claude-3.7-Sonnet achieves only 22.45% Pass@3 on the hard test split). Crucially, we demonstrate that SWE-Dev serves as an effective platform for model improvement: fine-tuning on training set enabled a 7B model comparable to GPT-4o on *hard* split, underscoring the value of its high-quality training data.

## 1 Introduction

Figure 1: Overview of SWE-Dev, a software development dataset providing feature development tasks with feature description and codebase as input and test cases for evaluation. It is uniquely grounded in real-world repositories and paired with executable test suites, enabling reliable, functionally verifiable supervision. SWE-Dev is evaluated on 37 autonomous coding systems and supports advanced training paradigms like SFT, RL, and multi-agent training.

Large Language Models (LLMs) are rapidly transforming autonomous programming, with capabilities extending from generating isolated code snippets to complex interactions within entire repositories GitHub (2021); Cursor (2023). As LLMs increasingly engage at this repository scale, rigorously evaluating their proficiency in handling complex coding systems becomes paramount for guiding their advancement. Current prominent benchmarks, while valuable, still struggle to judge how well LLMs perform in realistic, end-to-end development settings (Table 1). For example, SWE-Bench Jimenez et al. measures only localized bug fixes described by GitHub issues, and RepoBench Liu et al. evaluates the completion of a few unrelated functions within a repository. However, they overlook the core tasks of developing and integrating significant new functionalities, which truly define how real-world codebases evolve.

The task of developing and integrating new functionalities is defined as feature-driven software development, which consists of 40% coding tasks of all development efforts Xavier et al. (2017);

Harness Team (2025). Traditional feature development Coad et al. (1999); Palmer & Felsing (2002) typically involves intermediate processes such as developing an overall model or building feature lists. For coding systems, the critical way towards achieving more comprehensive and genuinely autonomous programming capabilities is on the final implementation of features. This end-to-end process, from interpreting requirements within large, existing codebases to generating functionally correct and integrated code, is what we refer to as end-to-end feature-driven software development.

Recognizing the central role of feature development and the limitations of current evaluation benchmarks, we introduce an feature-driven **S**oft**W**ar**E Dev**elopment dataset, **SWE-Dev**. which is the first large-scale dataset designed to evaluate and train autonomous AI systems on real-world end-to-end feature-driven software development tasks. It comprises 14,000 training and 500 test instances derived from over 1,000 open-source projects, and is distinguished by three key characteristics: (1) **Realistic scale and complexity**: SWE-Dev requires substantial code modifications (avg. 190 LOC across 3 files), challenging models with the cross-file dependencies, large contexts, and significant implementation scope characteristic of real-world feature development. (2) **Robust and grounded evaluation**: Each sample is grounded in a real open-source repository, guided by a well-defined project requirement description (PRD), and evaluated using executable test cases to ensure the satisfaction of the proposed implementation. This design ensures alignment between task objectives and evaluation, enabling robust assessment and model supervision. (3) **Verifiable training set with executable test suites**: Uniquely, all 14,000 training instances are paired with runnable environments and executable unit tests, providing crucial execution-based feedback that enables effective Supervised Fine-Tuning (SFT) validation, Reinforcement Learning (RL) with accurate rewards, and Multi-Agent System (MAS) training, refer to Table 1.

Our extensive experiments using SWE-Dev reveal several critical insights. Firstly, Repository-level feature development is challenging: our findings show even top-tier models like Claude-3.7-Sonnet Anthorpic (2025) and GPT-4o Hurst et al. (2024) solve only 22.45% *hard* samples and 68.70% *easy* samples with Pass@3. Secondly, MAS generally outperform single-agent baselines in modest margins. Interestingly, simple general-purpose MAS (e.g., Self-Refine Madaan et al. (2023b), Reflexion Shinn et al. (2023)) often outperform more complex code-specific agents, while requiring fewer model calls and lower cost. Lastly, task-specific training on this task gets substantial gains on all training methods. After training, a 7B fine-tuned model is comparable to GPT-4o on *hard* subset.

These findings point to several promising directions for future research. First, the difficulty of feature development for LLMs necessitates enhancing LLMs' core reasoning and long-context capabilities. Second, current MAS designs often suffer from unnecessary communication overhead and limited coordination efficiency. Future work should explore lightweight agent architectures and better context-sharing mechanisms for repository-level development. Lastly, our initial experiments with RL and role-based multi-agent training show that training can be beneficial, but headroom remains. Future work could investigate multi-agent training and long-context RL with SWE-Dev.

Our contributions are as follows:

- We introduce **SWE-Dev**, the first real-world dataset for autonomous end-to-end feature-driven software development. The dataset includes both training and test splits, each with runnable environments and test cases, enabling a wide range of evaluation and training.

- Our evaluations on SWE-Dev offer novel insights into the proficiency and deficiencies of **various coding systems (chatbot LLMs, reasoning LLMs, and MAS)** on feature development tasks.

- We demonstrate SWE-Dev **enabling and validating diverse training paradigms** (SFT, RL, and MAS training), establishing its utility for advancing training-based adaptation.

## 2 RELATED WORK

### 2.1 CODING BENCHMARKS

LLMs show significant potential in coding tasks, driving the need for robust benchmarks. Early benchmarks such as HumanEval Chen et al. (2021), MBPP Austin et al. (2021), APPS Hendrycks et al., and CodeContests Li et al. (2022) primarily focus on isolated, function-level tasks. These benchmarks test for correctness in constrained settings: short snippets, well-specified inputs, and short expected outputs. While useful for early-stage capability testing, such tasks fall short of reflecting the complex, multi-file dependency and long contexts nature of real-world software development tasks. To address this, repository-level benchmarks emerged, such as SWE-Bench Jimenez et al. (issue fixing), RepoBench Liu et al., and M2RC-Eval Liu et al. (2024) (code completion/understanding).

Table 1: Comparison of SWE-Dev with existing repository-level benchmarks. Task (FC: Function Completion, PG: Project Generation, LC: Line Completion, IS: Issue Solving, FD: Feature Development), usage of real-repository, availability of training sets, Number of Samples, and task statistics are compared here. Detailed statistics information is demonstrated e.g., line of code (LOC).

| | Task | Real Repo | Train Existence | w. Testcases | | # Samples | | Avg. PRD Tokens | Avg. LOC |
|---|---|---|---|---|---|---|---|---|---|
| | | | | Train | Test | Total | w. Testcases | | |
| **ComplexCodeEval** | FC | ✔ | ✗ | ✗ | ✗ | 7k | 0 | 134.2 | 38.21 |
| **CoderEval** | FC | ✔ | ✗ | ✗ | ✔ | 234 | 234 | 119.26 | 20.64 |
| **DevEval** | FC | ✔ | ✗ | ✗ | ✔ | 2k | 2k | 91.5 | 112 |
| **rSDE-Bench** | PG | ✗ | ✗ | ✗ | ✔ | 53 | 53 | 1553 | 157.88 |
| **M2rc-Eval** | LC | ✔ | ✔ | ✗ | ✗ | 59k | 0 | 0 | 1 |
| **RepoBench** | FC | ✔ | ✗ | ✗ | ✔ | 23k | 0 | 0 | 89 |
| **SWE-Bench** | IS | ✔ | ✔ | ✗ | ✔ | 19k | 2k | 195.1 | 32.8 |
| **SWE-Dev** | **FD** | ✔ | ✔ | ✔ | ✔ | 14.5k | **14.5k** | **1845.4** | **190** |

Despite this progress, they often face two main issues: (1) The scope of required code generation or modification remains limited (e.g., avg. 32.8 LOC in SWE-Bench, 38.21 LOC in ComplexCodeEval Li et al. (2024)), inadequately simulating large-scale feature development or refactoring. (2) Weak or inconsistent evaluation protocols: several benchmarks Feng et al. (2024); Liu et al. (2024); Liu et al. rely heavily on proxy metrics such as code similarity or static heuristics, which often fail to reflect functional correctness. This compromises both the robustness of evaluation and the comparability of results across models Lozhkov et al. (2024); Huang et al. (2022). SWE-Dev directly tackles these limitations by providing large-scale repository-level feature development tasks with executable unit tests. Its tasks involve substantial modifications, addressing shortcomings in both code scope and trainable environments, thereby significantly increasing task complexity and realism.

## 2.2 CODE LLMS TRAINING

Training LLMs for coding tasks typically involves three stages: pre-training, supervised fine-tuning (SFT), and reinforcement learning (RL). Pre-trained models such as StarCoder Lozhkov et al. (2024) and Phi Abdin et al. (2024a) leverage massive code corpora to learn syntax and general programming patterns. To improve instruction following and task completion, many works adopt SFT. Code Alpaca Wang et al. (2023b) employs self-instruct generation, WizardCoder Luo et al. leverages Evol-Instruct Xu et al. (2024) to synthesize more complex instructions. However, SFT fundamentally lacks exploration: it teaches models to imitate ground-truth outputs rather than to reason or build autonomously Ouyang et al. (2022). Beyond SFT, RL frameworks such as CodeRL Le et al. (2022) utilize test-case-based feedback to optimize model behavior. While promising, both SFT and RL approaches largely focused on function-level tasks, limiting their applicability to more complex development scenarios. To address this, SWE-Gym Pan et al. (2024) explores extending training to repository-scale tasks using multi-agent systems. However, due to the lack of an executable training set in SWE-Bench, SWE-Gym resorts to constructing a separate dataset of 2,438 tasks, ultimately yielding only 500 trajectory samples for training. In contrast, our proposed SWE-Dev provides a large-scale repository-level training set with runnable environments and unit-test-based supervision. It supports SFT, RL, and multi-agent training with executable feedback, enabling realistic and scalable development of code generation systems.

## 3 SWE-DEV

SWE-Dev is the first dataset designed to train and evaluate autonomous coding systems on feature-driven software development tasks. Each instance requires the model to implement a new capability within an existing codebase, based on a natural language requirement and evaluated through real-world unit tests. This section describes the construction of the dataset (§ 3.1), its core features (§ 3.2), and key statistics (§ 3.3).

## 3.1 DATASET CONSTRUCTION

Our dataset construction leverages a straightforward principle: test files in real-world repositories can serve both as a source of feature requirements and as a means of verifying correct implementation. In PyPI packages, developers create high-quality test files to ensure that specific modules or features function reliably across updates. For example, in `numpy`, `test_random.py` validates random array generation, aligning closely with the feature it tests. These test files provide executable, feature-specific validation, making them ideal for defining and evaluating development tasks.

Using these developer-authored tests as ground truth, we gain two advantages. First, they provide executable, functionality-level feedback for model evaluation. Second, by tracing the test cases back

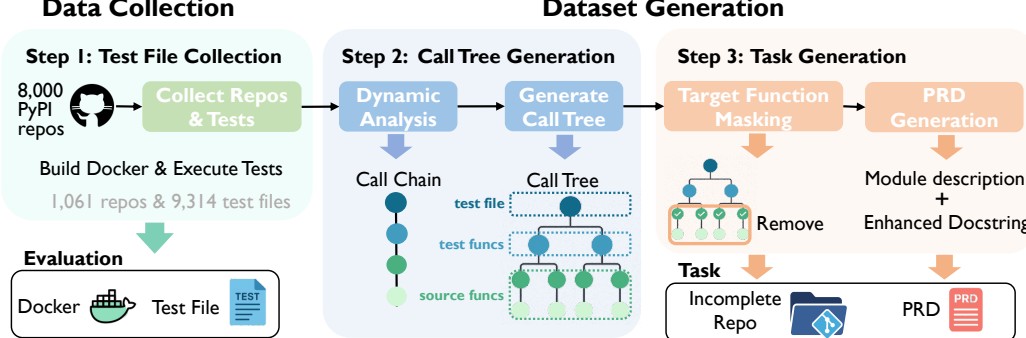

Figure 2: Overview of SWE-Dev dataset construction. **Step 1**: We collect real-world repositories with passing test files in Dockerized environments, **Step 2:** trace test executions to construct function-level call trees linking test cases to invoked source code, and **Step 3**: mask core functions while generating refined PRDs to create tasks. Each sample includes an incomplete repository, a natural language requirement, and executable test cases-enabling realistic, verifiable feature development.

to their associated implementation code, we can identify and mask the relevant source code, forming the basis of an incomplete development task. These traced functions also guide the generation of precise task descriptions. Based on this process, we divide our construction into three stages: (1) collecting repositories, test files and building Docker environments, (2) generating test-to-code call trees via dynamic tracing, and (3) creating the final task by masking the relevant source code and producing the feature specification.

**Step 1: Test File Collection**    To support realistic feature-level tasks and test-based evaluation, we begin by collecting real-world repositories that reflect common development scenarios. Specifically, we select 8,000 popular PyPI packages based on star counts. However, not all repositories are suitable: many lack usable tests or need sophisticated installation. Therefore, we applied a strict filtering process focused on test suite executability. Repositories were retained only if they met two criteria: (1) they include an identifiable test suite (e.g., using `pytest` or `unittest`), and (2) their test files could be executed successfully within the package Docker environment, with all tests passing. This ensures the resulting tasks are grounded in verifiable, runnable functionality. After filtering, we obtain 1,061 validated repositories (as of December 12, 2024) and 9,361 executable test files.

**Step 2: Call Tree Generation**    To locate the specific methods involved in implementing a feature, we capture the runtime interactions between test cases and their corresponding source code through dynamic analysis. This process has two main parts: (1) Dynamic analysis: We execute each test file using `pytest` inside a Docker environment and apply Python's built-in `trace` module to record all triggered functions in source code. This results multiple linear call chains that record the sequence of invoked source functions. (2) Call tree ensemble: We aggregate the call chains into into a hierarchical call tree, where the nodes of call tree represent functions, and edges capture dependency relationships. The call tree is rooted from test functions and followed by triggered source functions. The depth and width of the tree reflect the complexity of the feature logic, including nested structures and cross-file dependencies. These trees provide a precise mapping from test behavior to implementation code, enabling us to localize relevant functions and systematically control task difficulty later.

**Step 3: Task Generation**    Once we have localized the implementation logic using call trees, we transform it into a feature development task by (1) masking the existing implementations of feature components and (2) generating a natural language requirement for this feature. These components represent a typical development scenario where a new feature needs to be added to an existing system, but its implementation is yet to be realized. To achieve this, we perform the following: (1) Target function masking: We use structural properties of the call tree (e.g., depth and node count) to identify key function nodes that represent the core logic for the feature. The corresponding implementation code is removed from the repository, creating a space for the new feature to be integrated. (2) Project Requirement Document (PRD) generation: We construct the feature description in PRD by using GPT-4o to synthesize a high-level module description from the test file and augmenting the masked function's docstring with implementation-level details. These two elements are combined into PRD, which serves as the task prompt. See example in Fig. 9 and prompts in Appendix J.

## 3.2 DATASET FEATURES

**Controlled Complexity via Call Tree:** Leveraging call-tree analysis, SWE-Dev enables systematic adjustment of task difficulty by adjusting dependency depth for task generation. This uniquely supports rigorous assessment of model capabilities against varying complexities, see § 5 discussion.

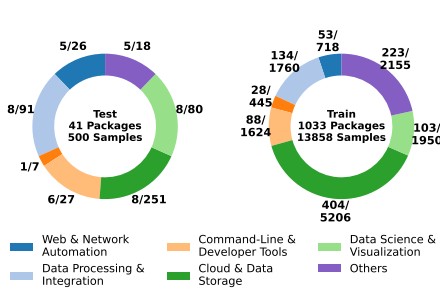

Figure 3: Distribution of SWE-Dev training and test samples across 6 major PyPI application domains.

Table 2: Basic statistics of SWE-Dev, including task specification length, repository scale, ground truth implementation size, and evaluation test coverage for both train and test splits.

| Category | Metric | Test | | Train |
|---|---|---|---|---|
| | | Easy | Hard | |
| **Size** | # Samples | 250 | 250 | 14000 |
| | # Total repos | 39 | 39 | 1033 |
| **Task** | # Avg. tokens | 1499 | 2148 | 1833 |
| **Codebase** | # Avg. Files | 71.31 | | 64.40 |
| | # Avg. LOC | 21308 | | 20206 |
| **GT Code** | # Avg. LOC | 109.1 | 172.4 | 199.92 |
| | # Avg. funcs | 4.75 | 6.972 | 6.03 |
| **Tests** | # Avg. test lines | 134.8 | 123.9 | 90.9 |
| | # Avg. testcases | 6.62 | 4.29 | 5.92 |

**Reliable Test-Based Evaluation:** Assessment uses the original, developer-authored unit tests, validated for correct execution in a controlled environment. This execution-based pass/fail verification provides an objective, reproducible, and functionally accurate measure of code, directly reflecting real-world correctness criteria.

**Executable Training Support:** SWE-Dev includes runnable environments and test cases for every sample, enabling training paradigms such as SFT and RL with execution-based feedback.

### 3.3 STATISTICS

Table 2 summarizes the key statistics of SWE-Dev, which consists of 14,000 training and 500 test samples. The test set is manually curated and split into two difficulty levels: *easy* and *hard* (250 instances each). Each dataset instance comprises four components: (1) the task, specified by a PRD, with its token count reflecting instruction length; (2) the codebase, referring to the non-test portion of the repository, where we report the number of files and lines of code (LOC); (3) the ground truth (GT) code to be implemented, measured by its LOC and number of target functions; and (4) the test suite, evaluated via the number of test cases and total test LOC per sample. Figure 3 shows the distribution of training and test samples across six major PyPI application domains, demonstrating the diversity of software categories represented in the dataset. More statistics are in Appendix C.

### 4 EXPERIMENT

In this section, we empirically evaluate the effectiveness of various coding systems and training paradigms on SWE-Dev. We first compare the performance of single-LLM(§ 4.1) and MAS(§ 4.1.2) on the feature-driven software development tasks. Then, the effectiveness of different training approaches, including SFT (§ 4.2.1), RL (§ 4.2.2), and multi-agent training (§ 4.2.3) is discussed.

**Setup.** We employed the Pass@$k$ as an evaluation metric in SWE-Dev Chen et al. (2021). For inference code context, since SWE-Dev requires both the PRD and codebase as inputs. The codebases consist of many tokens (an average of 202K lines, see Table 2), exceeding typical LLM context window. Thus, in all the experiments below, we provide only the relevant code context—i.e., the specific files involved in the task—rather than the entire codebase.

### 4.1 TESTING RESULTS

This section presents the performance of 17 chatbot LLMs, 10 reasoning LLMs, and 10 multi-agent systems on SWE-Dev, under the single-LLM and multi-agent settings. Full details of the evaluated methods are provided in Appendix H.1.

### 4.1.1 SINGLE LLM INFERENCE

**SWE-Dev presents substantial challenges for current LLMs, revealing a clear gap between existing coding capabilities and real-world software engineering demands.** Figure 4 reports Pass@3 performance of chatbot and reasoning LLMs on SWE-Dev. We observe that:

(1) LLMs perform better on the *easy* split than the *hard* split. (2) Performance generally scales with model size, especially for LLMs within the same family, aligning with our understanding of LLM capabilities. (3) Even the best-performing LLM (Claude-3.7-Sonnet Anthorpic (2025)) achieves just over 20% on the *Hard* split. This still falls short of achieving strong performance, indicating

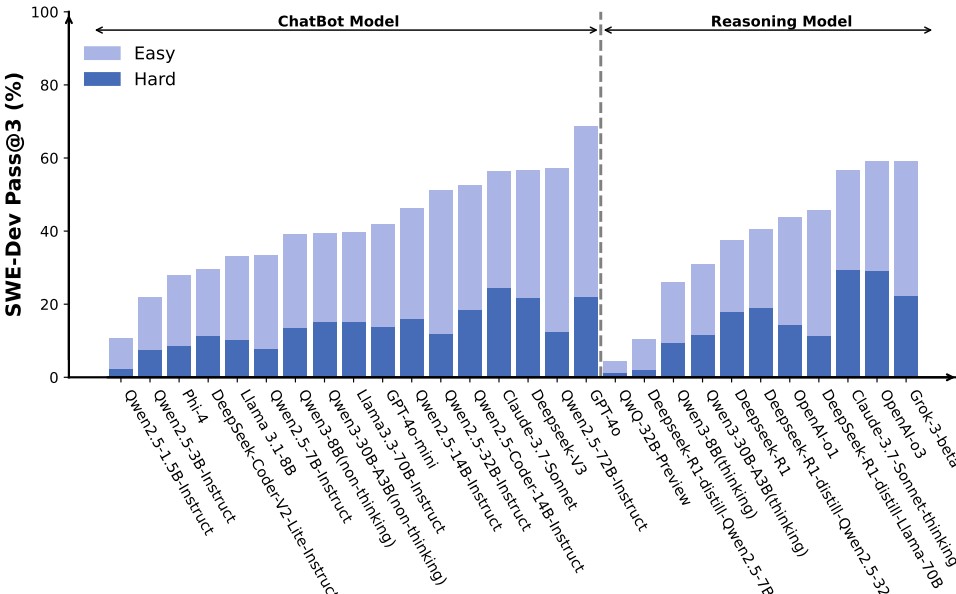

Figure 4: Comparison of Pass@3 scores for 17 chatbot and 10 reasoning LLMs on SWE-Dev across *Easy* and *Hard* splits. SWE-Dev poses substantial challenges and effectively distinguishes model capabilities under both difficulty levels. See Appendix D for full results.

that current models are not yet fully capable of handling tasks that approximate the complexity of real-world development scenarios.

**Reasoning models generally underperform their base counterparts, with Claude-3.7-Sonnet being a exception.** While Claude with thinking outperforms its base variant, most reasoning models yield worse results. This suggests that current reasoning strategies do not consistently translate into gains for complex, repository-level generation tasks. We further explain this in Appendix E.2.

**SWE-Dev provides stable and discriminative evaluation of model capabilities.** Figure 5 compares the performance of Qwen2.5 Qwen et al. (2025a) family on SWE-Dev, HumanEval Chen et al. (2021), and ComplexCodeEval Feng et al. (2024) across three runs. We use Pass@1 for SWE-Dev and HumanEval and ComplexCodeEval for CodeBLEU Ren et al. (2020). The lines represent the average performance, and the shaded regions show the variance. We observe that: (1) SWE-Dev yields low variance performance and consistent scaling with model size, demonstrating SWE-Dev's stability and reliability in evaluating model capabilities. (2) In contrast, HumanEval—despite being stable—is too simple to differentiate models meaningfully. (3) Meanwhile, ComplexCodeEval shows high variance due to its reliance on similarity-based metrics, CodeBLEU, which limits its reliability for evaluating complex generation tasks.

### 4.1.2 MULTI-AGENT INFERENCE

Table 3 compares the performance, call times and total costs of various MAS against the single-agent baseline driven by GPT-4o-mini. Details of MAS are given in AppendixH.1. Key observations are

**MAS generally outperforms single-agent baselines on complex tasks.** While the single-agent approach achieves only 11.09% Pass@1 on hard tasks, Self Refine Madaan et al. (2023a) and EvoMAC Hu et al. (2025b) improve performance to 20.03% and 13.60%, respectively. These results highlight the advantage of MAS in solving complex, reasoning-intensive problems.

**Simpler multi-agent strategies offer strong performance–efficiency trade-offs.** Methods such as Self Refine strike an effective balance between performance and cost. On the easy subset, Self Refine achieves the highest Pass@1 of 40.02% using only 5 calls. In contrast, more complex systems like ChatDev, despite making over 26 calls, fall behind in performance (35.13%), indicating that additional agent complexity does not necessarily lead to better results.

**Human-designed, workflow-heavy MAS often introduce unnecessary overhead.** Systems with manually defined roles and interaction protocols, such as ChatDev and MapCoder, tend to be less effective. On hard tasks, ChatDev requires over 30 calls yet only achieves 11.7%, while Map-Coder performs even worse, with 5.87% despite 23.41 calls. These results suggest that handcrafted workflows may introduce redundant operations without improving code generation quality.

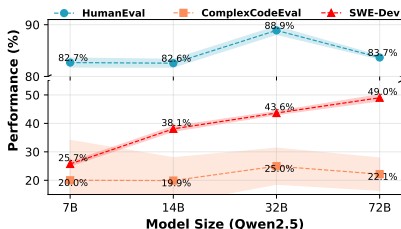

Figure 5: Comparison of benchmarks on various model sizes. SWE-Dev shows clear performance scaling with model size, while HumanEval Chen et al. (2021) fails to distinguish between models. ComplexCodeEval Feng et al. (2024) using CodeBLEU Ren et al. (2020) exhibits high variance, making it less stable for evaluation.

Table 3: Comparison of general and code-specific MAS on SWE-Dev driven by GPT-4o-mini. **Bold** highlights the best performance; underlined indicates results worse than the single-agent baseline. Most MAS methods outperform the single agent, and simpler general MASs are more effective and efficient than complex coding-specific MASs.

| Method | Easy | | | Hard | | |
|---|---|---|---|---|---|---|
| | Pass@1 | Calls | Price($) | Pass@1 | Calls | Price($) |
| **Single** | 34.47 | 1.00 | 0.75 | 11.09 | 1.00 | 0.97 |
| **Reflexion** | 39.77 | 2.12 | 0.83 | 13.32 | 2.18 | 1.35 |
| **Self Refine** | **40.02** | 5.00 | 5.78 | **20.03** | 5.00 | 5.8 |
| **Self Consistency** | 37.62 | 6.00 | 4.30 | 18.55 | 6.00 | 7.08 |
| **LLM Debate** | 38.48 | 7.00 | 5.95 | 14.56 | 7.00 | 9.35 |
| **MAD** | 31.50 | 7.00 | 2.48 | 15.31 | 7.00 | 3.40 |
| **Agentverse** | **38.67** | 4.52 | 1.40 | 13.42 | 4.83 | 2.90 |
| **EvoMAC** | 34.59 | 7.98 | 3.20 | **13.60** | 8.30 | 4.65 |
| **MetaGPT** | 29.56 | 9.69 | 2.20 | 9.25 | 10.37 | 4.95 |
| **MapCoder** | 24.55 | 21.01 | 6.05 | 5.87 | 23.41 | 10.55 |
| **ChatDev** | 35.13 | 26.61 | 3.53 | 11.70 | 30.87 | 6.10 |

Our results highlight MAS's potential for complex tasks on SWE-Dev but reveal a gap between simple and complex MAS, indicating that scalable, efficient MAS remain a challenge. Future work could focus on balancing collaboration benefits with resource costs and mitigating error amplification from LLM hallucinations across agent interactions.

## 4.2 TRAINING RESULTS

In this section, we evaluate SWE-Dev's support for different training methods, including SFT, RL. Additionally, we present preliminary results from our exploration of multi-agent training, offering an initial assessment of MAS-based learning. For detailed training setups, refer to the Appendix H.2.

### 4.2.1 SINGLE LLM SFT

We conducted experiments on Qwen2.5-Intstruct models of various sizes (0.5B, 1.5B, 3B, and 7B) to assess the impact of SFT on performance in SWE-Dev. Experimental setting is in Appendix H.3.

**Training significantly improves performance across model sizes.** SFT leads to substantial performance improvements across all model sizes, especially for harder tasks. As shown in Table 5, the 7B model achieves a Pass@1 of 36.90% on the easy task set after fine-tuning, up from 25.74% in the zero-shot setting. On the hard task set, the Pass@1 increases from 6.68% to 18.89%, demonstrating the clear benefits of training in enhancing model performance.

**SWE-Dev effectively supports the scaling law of training.** Figure 6 illustrates the scaling law of training using Qwen2.5-7b-instruct. In this experiment, we measured model performance across varying amounts of fine-tuning data, specifically tracking changes in Pass@1 for both easy and hard task. As shown in the figure, performance improves steadily as the amount of fine-tuning data increases, with larger improvements observed for harder tasks.

In summary, our results underscore the importance of fine-tuning in improving performance on SWE-Dev. The scaling law observed here further supports the idea that SWE-Dev is a valuable dataset for studying the effects of model size and training data on task performance.

### 4.2.2 SINGLE LLM RL

SWE-Dev provides precise test cases enabling accurate rewards for coding tasks, supporting both online and offline RL. In this section, we explore the impact of RL on the Qwen2.5-7B-instruct using SWE-Dev. Considering the computational cost of RL, we limit our experiments in this section to 2k training samples. For full training setup, refer to the Appendix H.4.

**Both online and offline RL improve performance, especially on hard tasks.** Table 4 shows that both PPO Schulman et al. (2017) and DPO Rafailov et al. (2023) significantly improve Pass@1 performance, especially on the *Hard*

Table 4: Performance comparison of Qwen2.5-7B-Instruct as base model, SFT-Tuned and RL-Tuned models on SWE-Dev.

| Method | Pass@1 | | Pass@3 | |
|---|---|---|---|---|
| | Easy | Hard | Easy | Hard |
| **vanilla** | 25.74 | 6.68 | 33.35 | 7.73 |
| **SFT** | 27.09 | 9.77 | **34.49** | 13.63 |
| **PPO (online RL)** | **28.30** | **12.25** | 32.69 | 14.33 |
| **DPO (offline RL)** | 25.89 | 10.36 | 31.32 | **14.66** |

Table 5: Comparison of zero-shot and SFT performance (Pass@1) on SWE-Dev using Qwen2.5 models. Results are reported on both Easy and Hard test splits across model sizes from 0.5B to 7B. The Δ columns indicate relative improvement after SFT. Fine-tuning yields consistent gains.

|  | **Zero-shot** |  | **SFT** |  |  |  |
|---|---|---|---|---|---|---|
|  | Easy | Hard | Easy | Δ(%) | Hard | Δ(%) |
| **0.5B** | 6.39 | 1.00 | 12.12 | +90 | 4.37 | +337 |
| **1.5B** | 8.05 | 1.23 | 18.20 | +126 | 7.64 | +521 |
| **3B** | 15.93 | 5.27 | 27.53 | +73 | 12.46 | +136 |
| **7B** | 25.74 | 6.68 | 36.90 | +43 | 18.89 | +183 |

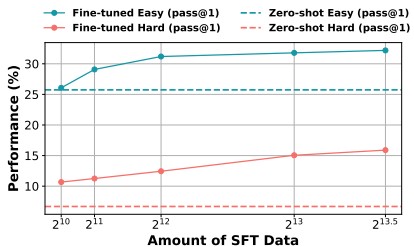

Figure 6: Training data scaling of SFT Qwen2.5-7B-instruct on SWE-Dev. As data size increases, performance improves steadily under SFT.

split. Furthermore, PPO outperforms SFT on the same training data. These findings highlight the advantages of RL training.

**RL boosts one-shot success but not multi-sample gains.** While RL fine-tuning yields improvements in Pass@1, it has minimal impact on Pass@3. Specifically, PPO achieves a Pass@1 of 28.30% on easy tasks, a noticeable increase from the base model's 25.74%, but the Pass@3 remains lower than the SFT-training, even the original model's performance. These results suggest that RL can be beneficial in refining Pass@1, particularly for more complex tasks, by increasing the model's efficiency in generating correct answers in fewer attempts. However, this efficiency comes at the cost of reduced exploration. This aligns with findings from prior work Yue et al. (2025). Therefore, while RL improves performance, significant headroom remains, and more advanced methods or further training are needed to achieve improvements across tasks.

### 4.2.3 MULTI-AGENT TRAINING

MAS has shown promising results on SWE-Dev, and we further investigate the training process of MAS on this dataset. As depicted in Fig. 7, the ground truth test case supervision in SWE-Dev enables EvoMAC Hu et al. (2025b) to improve its performance across multiple rounds of reasoning. This iterative refinement process motivates us to explore EvoMAC as the MAS for training in SWE-Dev. We apply rejection sampling to enhance agent performance via role-wise training.

**Trajectory Collection.** We use Qwen2.5-7B-Instruct to collect training data for the MAS, following these steps: (1) **EvoMAC iterative reasoning**: EvoMAC performs multiple reasoning rounds, benefiting from ground truth test case supervision to progressively improve its performance. (2) **Rejection sampling**: At each iteration, we apply rejection sampling based on training sample testcases to select high-quality trajectories that show improvement over the previous round, ensuring the retention of beneficial data. (3) **Role-wise training**: The selected trajectories are used to role-wise train two agents (organizer and coder) in EvoMAC, allowing each agent to specialize in its task for better overall performance.

**Training Effectiveness.** Table 6 presents the performance of different training configurations in terms of Pass@1. We see that: i) Fine-tuning both the organizer and coder agents results in the highest performance, with Pass@1 of 31.65% on easy tasks and 12.70% on hard tasks, outperforming all other configurations; ii) When only one agent is fine-tuned, we also see improvements over the baseline. These findings highlight the effectiveness of role-wise training for MAS training.

## 5 DATASET ANALYSIS

We analyze SWE-Dev's task complexity, evaluation setup, and PRD quality to demonstrate its uniqueness and reliability.

**Analysis of Task Difficulty and Call Tree Characteristics.** We analyze how task difficulty in SWE-Dev correlates with call tree complexity. As introduced in § 3.1, a call tree reflects the dynamic function invocation structure for this task, where nodes represent functions and edges denote their call relationships. We use two metrics: depth, indicating the maximum call nesting, and node count, representing the total number of distinct functions involved in the task. Fig. 8a shows that GPT-4o's performance declines as depth and node count increase, revealing a strong correlation between structural complexity and task difficulty. This suggests that deeper and broader call structures introduce more functional requirements and interdependencies, making tasks more challenging.

**Evaluation Method Precision.** SWE-Dev uses execution-based evaluation with test cases, enabling precise performance signals. We compare metrics: Exact Match (EM) Liu et al. (2024), Exact Sequence (ES) Liu et al. (2024), CodeBLEU Ren et al. (2020), and Pass@3, using Qwen2.5 models

Table 6: Comparison of multi-agent role-wise training, base MAS and single LLM's performance on Qwen2.5-7B-Instruct. $\Delta$ indicates the relative improvement over the base MAS system. Partial fine-tuning of either agent also leads to consistent gains, demonstrating the effectiveness of role-specific supervision enabled by SWE-Dev.

|  | Org | Coder | Easy | $\Delta$(%) | Hard | $\Delta$(%) |
|---|---|---|---|---|---|---|
| **Single** | - | - | 25.74 | - | 6.68 | - |
| **MAS** | base | base | 26.64 | - | 7.39 | - |
|  | FT | base | 30.04 | +12.76 | 12.36 | +67.25 |
|  | base | FT | 31.42 | +17.94 | 11.49 | +55.48 |
|  | FT | FT | **31.65** | +18.80 | **12.70** | +71.85 |

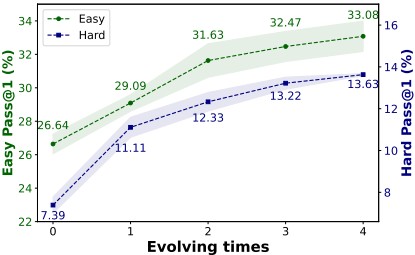

Figure 7: EvoMAC performance trajectory under ground truth test case supervision on SWE-Dev with Qwen2.5-7B-Instruct. Evo-MAC iteratively improves across reasoning rounds, guided by executable test feedback.

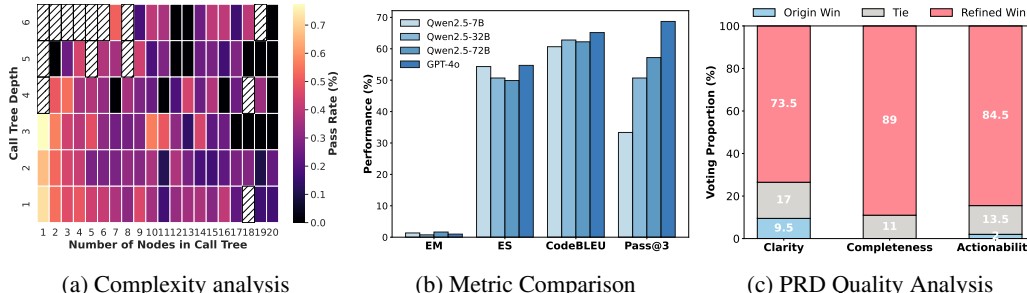

(a) Complexity analysis  (b) Metric Comparison  (c) PRD Quality Analysis

Figure 8: Analysis of SWE-Dev Benchmark Characteristics. (a) Compares GPT-4o's performance across tasks grouped by call tree depth and node count, showing that greater structural complexity correlates with lower accuracy. (b) Compares several evaluation metrics; Pass@3 shows the clearest differentiation across model scales. (c) Compares human ratings of original vs. refined PRD on 100 samples of 3 dimensions, revealing SWE-Dev's high PRD quality.

and GPT-4o. As Fig 8b shows, Pass@3 best reflects capability scaling, separating models by size and quality. In contrast, EM, ES, and CodeBLEU show minimal variance, failing to distinguish models. This demonstrates that SWE-Dev's test-case-based evaluation provides a more robust and realistic signal of model performance, better reflecting the functional correctness required in real-world software development.

**PRD Quality.** SWE-Dev includes a PRD for each task to simulate realistic developer-facing requirements, which are primarily derived from the original docstrings found within the repository source code. While many functions in open-source code include docstrings, we found that these are often incomplete—lacking clear descriptions of behavior, parameters, or edge cases. To improve instruction clarity without fabricating content, we lightly refine existing docstrings using GPT-4o, grounded in the related file and surrounding context. To evaluate instruction quality, we conducted a human assessment on 100 sampled tasks. Two experienced engineers rated the original and refined PRDs along Actionability, Completeness, and Clarity (Appendix E.1 includes human instruction). As shown in Fig. 8c, refined PRDs consistently scored higher across all dimensions. This supports SWE-Dev's goal of providing realistic, well-scoped requirements for reliable model evaluation.

## 6 CONCLUSION

In this work, we introduced SWE-Dev, the first dataset for evaluating and training autonomous coding systems on end-to-end feature-driven development task. SWE-Dev consists of 14,000 training and 500 test instances, each uniquely equipped with runnable environments and developer-authored executable unit tests, which provides essential execution-based feedback for advanced training paradigms like SFT, RL, and multi-agent learning. Our experiments show feature-driven software development is profoundly challenging for current autonomous coding systems. We also validate that training on SWE-Dev can yield encouraging performance gains. These findings validate SWE-Dev as a critical platform for identifying current limitations and fostering breakthroughs in AI-driven software development. We hope the release of SWE-Dev spurs innovation in long-context reasoning, agent orchestration, and execution-aware training towards genuinely autonomous software engineering.

ETHICS STATEMENT

SWE-Dev is collected entirely from publicly available repositories with open-source licenses that permit the usage of software in accordance with our contributions. The repositories included in the SWE-Dev train split does not contain any GPL-licensed or unlicensed code. The distribution of licenses in the dataset is as follows: MIT (697 repositories), Apache-2.0 (165 repositories), and BSD 2/3-Clause (148 repositories). SWE-Dev's repository selection process does not rely on biased or discriminative heuristics. Repositories are chosen based on popularity metrics, and this selection process does not implicitly or explicitly favor any particular type of project or developer. For the dataset release, we make the SWE-Dev task instances, the collection and evaluation infrastructure, and the experimental results publicly available. To facilitate feedback and improvements, we will provide open channels for communication.

REPRODUCIBILITY STATEMENT

Our code is available here https://anonymous.4open.science/r/SWE-Dev. The training parameters and detailed experimental setup are described in Appendix H. Our approach and experiments were designed with reproducibility in mind, and we encourage others to build upon our work using the provided resources.

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

APPENDIX

# A    LLM USAGE DISCLOSURE

In our work, we utilized GPT-4o exclusively to enhance readability and improve language fluency during the writing process. We take full responsibility for the content of this publication, including any content generated by LLM.

# B    OTHER RELATED WORK

**RL for Software Engineering**

Earlier RL and RLHF work on code has mostly targeted short, function-level problems (e.g., HumanEval-style benchmarks), where the model produces a single solution in one shot with an immediate test-based reward. More recently, several works have applied RL to more complex SWE settings Golubev et al. (2025). SWE-RL Wei et al. (2025) uses policy-gradient RL in an agentless scaffold on SWE-Bench, but still frames the task as single-turn patch generation, which sidesteps the challenges of stateful, multi-step interaction and long-horizon credit assignment. In contrast, DeepSWE Luo et al. (2025), and SkyRL-v0 Cao et al. (2025) scale RL to interactive, multi-turn SWE agents operating over full repositories, tools, and long contexts. Our work is complementary: rather than proposing a new agent or RL algorithm, SWE-Dev provides a large, PRD-driven, feature-level benchmark with executable rewards and examines how standard SFT and PPO behave on multi-file feature implementation. This makes SWE-Dev a natural substrate for future RL-based SWE agents in the spirit of these recent works.

# C    DATASET

## C.1    DATASET INFORMATION

Figure 9 illustrates a typical task instance in the SWE-Dev, detailing the entire development workflow. The process begins with the Project Requirement Description (PRD), which provides instructions and specifies features to be implemented. Methods to be evaluated then generate code to complete the features mentioned in the PRD, which is subsequently verified against the test suite to produce pass/fail results to calculate pass rate. Additionally, the ground truth implementation for each PRD is included for reference. The tasks in SWE-Dev simulate real-world software development cycles within a repository context. For detailed information about each data field included in SWE-Dev tasks, please refer to Table 7.

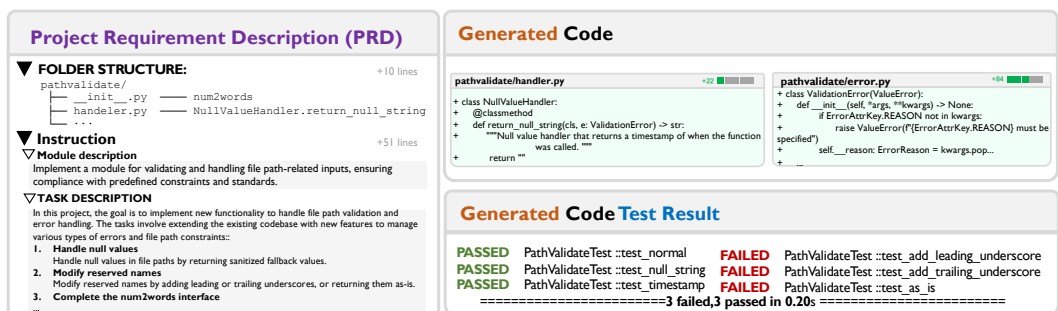

Figure 9: Example of a SWE-Dev Sample. Each sample includes a Project Requirement Description (PRD) with folder structure, module-level task description, and masked docstrings; the corresponding ground truth implementation. Generated code is evaluated by the test function execution results. This structure supports realistic, testable feature development in a repository context.

Table 7: Description of each field of an SWE-Dev task instance object.

| Field | Description |
|---|---|
| PRD | A natural language document describing the project requirements, including the specific features to be implemented. |
| file_code | Incomplete code contents of the core files involved in the task. |
| test_code | Content of the test code used to verify the task's functionality. |
| dir_path | Root directory path of the project corresponding to this task instance. |
| package_name | Name of the software package or module to which this task instance belongs. |
| sample_name | Unique identifier or name for this task instance or sample within the benchmark. |
| src_dir | Relative directory path where the source code files for the project or task are located. |
| test_dir | Relative directory path where the test code files for the project or task are located. |
| test_file | Relative path of the unit test file used for executing tests. |
| GT_file_code | Ground Truth source code for the file to complete. |
| GT_src_dict | Ground Truth source dictionary, mapping file names/paths to their expected correct code content. |
| dependency_dict | Dictionary describing the dependencies required by the current task (e.g., internal modules) and their relationships. |
| call_tree | Function call tree or call graph of the code, representing the relationships between function calls. |

## C.2 DATASET DISTRIBUTION

We present the distribution statistics of the training and test sets in SWE-Dev. Each sample includes a Project Requirement Document (PRD), which describes the feature to be implemented. The average PRD length is 1,845.4 tokens. On average, each sample includes at least 5 unit tests for functional evaluation, spans 3 source files, and requires the implementation of approximately 6 functions.

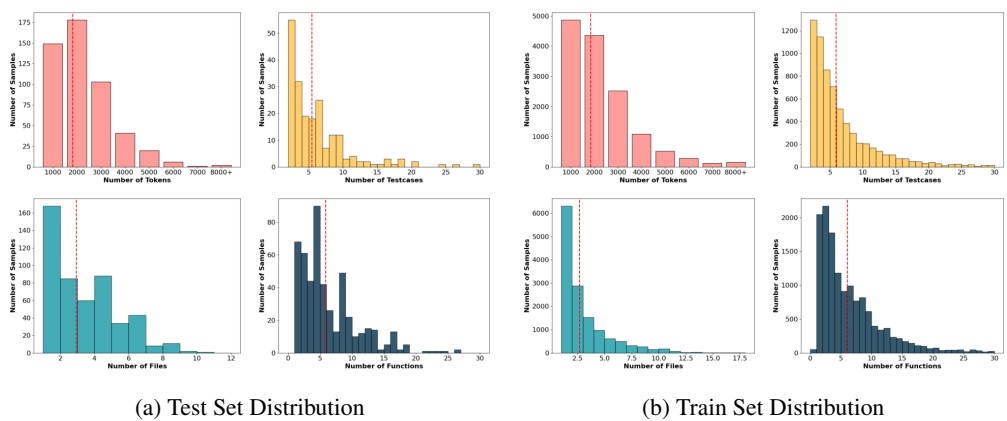

(a) Test Set Distribution (b) Train Set Distribution

Figure 10: Dataset distribution of PRD tokens, number of testcases, number of files to complete, and number of functions per sample.

## C.3 DATASET DIVERSITY

We assess the diversity of SWE-Dev from two perspectives: sample-level diversity (Figure 11 and Figure 12), and package-level diversity (Figure 26).

**Sample Diversity via t-SNE.** To visualize the diversity of feature requirements, we perform t-SNE on PRD embeddings generated using OpenAI's text-embedding-ada-002 model.[1] We use 500 test samples and randomly sample 2,000 training samples. Each point represents a PRD, and the color denotes its corresponding package. The resulting distribution reveals rich semantic variation across tasks, even within the same package, highlighting the dataset's diversity in both content and functionality.

**Package Category Diversity.** To analyze the functional diversity of the dataset, we classify packages into high-level categories based on their primary domain (e.g., web development, data science, utilities). The classification is performed using GPT-4o-mini with the prompt provided below (see Figure 26). The resulting distribution confirms that SWE-Dev spans a broad spectrum of software domains.

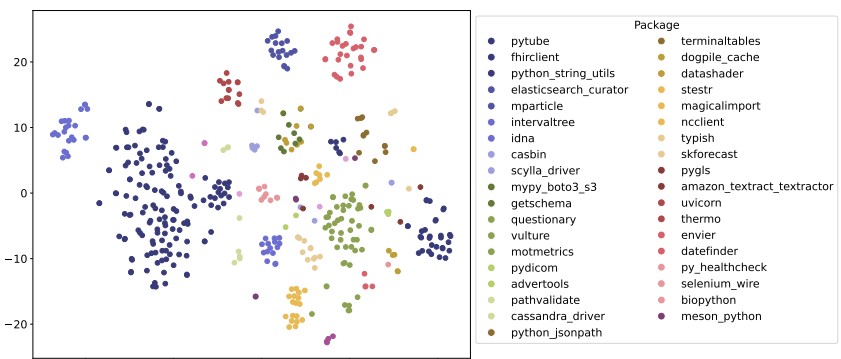

Figure 11: t-SNE visualization of PRD in test set

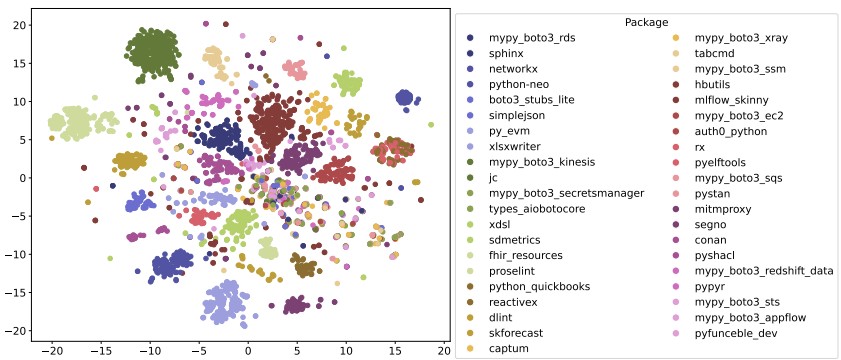

Figure 12: t-SNE visualization of PRD in train set

# D INFERENCE RESULTS

To assess the capabilities and limitations of current LLMs on realistic feature-driven software development tasks, we conduct comprehensive inference-time evaluations on SWE-Dev. We study both single-agent and multi-agent systems, measuring their performance under consistent execution-based evaluation.

## D.1 SINGLE-AGENT LLM PERFORMANCE

We evaluate 27 state-of-the-art LLMs, including general-purpose chatbot models (e.g., GPT-4o, Claude 3.7) and reasoning models, shown in Table 9. Models are assessed using Pass@1 and Pass@3 on SWE-Dev's test set. To contextualize benchmark difficulty, we also compare results

---

[1] https://platform.openai.com/docs/guides/embeddings

on HumanEval Chen et al. (2021) and ComplexCodeEval Feng et al. (2024), using Pass@3 and CodeBLEU respectively. Our findings show that SWE-Dev poses significantly greater challenges than existing benchmarks, with leading models achieving under 25% Pass@3 on hard tasks (Table 8).

Table 8: Evaluation of model performance across benchmarks. This table compares 37 general-purpose and reasoning-focused LLMs on SWE-Dev (Pass@1 and Pass@3 for Easy and Hard splits), ComplexCodeEval (CodeBLEU), and HumanEval (Pass@1 and Pass@3).

| | Model | SWE-Dev pass@1 Easy | Hard | SWE-Dev pass@3 Easy | Hard | ComplexCodeEval CodeBLEU | HumanEval pass@1 | HumanEval pass@3 |
|---|---|---|---|---|---|---|---|---|
| Chatbot | Qwen2.5-1.5B-Instruct | 8.05% | 1.23% | 10.76% | 2.22% | 29.72% | 57.20% | 69.76% |
| | Qwen2.5-3B-Instruct | 15.93% | 5.27% | 21.99% | 7.47% | 12.27% | 62.68% | 75.00% |
| | Qwen2.5-7B-Instruct | 25.74% | 6.68% | 33.35% | 7.73% | 20.00% | 82.68% | 87.13% |
| | Llama-3.1-8B-Instruct | 26.43% | 7.94% | 33.01% | 10.24% | 20.18% | 68.20% | 77.87% |
| | Qwen3-8B | 34.04% | 12.09% | 39.26% | 13.33% | 17.47% | 86.34% | 89.09% |
| | Qwen2.5-Coder-14B-Instruct | 39.51% | 14.82% | 52.49% | 18.44% | 35.52% | 90.48% | 92.93% |
| | Qwen2.5-14B-Instruct | 38.08% | 13.16% | 46.32% | 15.89% | 19.90% | 82.56% | 87.87% |
| | DeepSeek-Coder-V2-Lite-Instruct | 21.53% | 8.19% | 29.68% | 11.33% | 26.63% | 80.98% | 85.18% |
| | Qwen3-30B-A3B | 35.84% | 12.76% | 39.45% | 15.20% | 14.84% | 89.27% | 90.30% |
| | Phi-4 | 21.99% | 5.57% | 27.89% | 8.56% | 33.85% | 86.46% | 90.01% |
| | Qwen2.5-32B-Instruct | 43.64% | 10.15% | 51.24% | 11.69% | 19.76% | 88.90% | 92.44% |
| | Qwen2.5-72B-Instruct | 49.01% | 10.62% | 57.20% | 12.33% | 22.15% | 83.66% | 86.46% |
| | Llama3.3-70B-Instrcut | 33.84% | 12.85% | 39.57% | 14.95% | 21.29% | 84.51% | 88.54% |
| | Deepseek-V3 | 41.95% | 16.22% | 56.79% | 21.62% | 28.32% | 90.36% | 92.92% |
| | GPT-4o | **54.37%** | 19.13% | **68.70%** | 21.91% | 33.38% | 88.41% | 92.93% |
| | GPT-4o-mini | 34.47% | 11.09% | 41.94% | 13.84% | 25.00% | 85.97% | 89.00% |
| | Claude-3.7-Sonnet | 53.09% | 19.74% | 56.35% | 24.25% | 29.63% | 93.66% | 95.36% |
| Reasoning | Claude-3.7-Sonnet-thinking | 49.47% | **22.51%** | 56.58% | **29.28%** | **29.80%** | 91.22% | 97.62% |
| | Deepseek-R1-distill-Qwen2.5-7B | 6.30% | 1.29% | 10.30% | 1.95% | 21.05% | 86.10% | 93.29% |
| | Qwen3-8B-thinking | 19.47% | 6.36% | 25.91% | 9.22% | 20.98% | 89.63% | 91.89% |
| | Qwen3-30B-A3B-thinking | 23.63% | 8.30% | 31.00% | 11.60% | 25.00% | 93.04% | 99.57% |
| | Deepseek-R1-distill-Qwen2.5-32B | 24.25% | 9.79% | 40.53% | 19.04% | 27.98% | 95.17% | 97.87% |
| | DeepSeek-R1-distill-Llama-70B | 32.73% | 8.19% | 45.72% | 11.33% | 25.95% | 96.95% | 98.53% |
| | Deepseek-R1-671B | 28.55% | 12.84% | 37.62% | 17.72% | 34.47% | **98.65%** | **100%** |
| | QwQ-32B-Preview | 4.50% | 0.70% | 8.90% | 1.22% | 24.78% | 82.31% | 97.01% |
| | grok-3-beta | 53.63% | 18.97% | 59.08% | 22.26% | 27.96% | 87.15% | 89.99% |
| | o1 | 36.36% | 11.09% | 43.77% | 14.27% | 33.63% | 97.43% | 98.78% |
| | o3 | 51.21% | 21.86% | 59.05% | 28.98% | 26.53% | 98.04% | 98.78% |

Table 9: Information of evaluated LLMs.

| Model | Size | Release Date | Open | Link |
|---|---|---|---|---|
| Qwen/Qwen2.5-Coder-14B-Instruct | 14B | 2024-11-12 | ✓ | Qwen2.5-Coder-14B-Instruct |
| deepseek-ai/DeepSeek-Coder-V2-Lite-Instruct | 16B | 2024-06-17 | ✓ | DeepSeek-Coder-V2-Lite-Instruct |
| microsoft/phi-4 | 14B | 2024-12-12 | ✓ | phi-4 |
| Qwen/Qwen2.5-1.5B-Instruct | 1.5B | 2024-09-19 | ✓ | Qwen2.5-1.5B-Instruct |
| Qwen/Qwen2.5-3B-Instruct | 3B | 2024-09-19 | ✓ | Qwen2.5-3B-Instruct |
| Qwen/Qwen2.5-7B-Instruct | 7B | 2024-09-19 | ✓ | Qwen2.5-7B-Instruct |
| meta-llama/Llama-3.1-8B-Instruct | 8B | 2024-07-23 | ✓ | Llama-3.1-8B-Instruct |
| Qwen/Qwen3-8B | 8B | 2025-04-29 | ✓ | Qwen3-8B |
| Qwen/Qwen2.5-14B-Instruct | 14B | 2024-09-19 | ✓ | Qwen2.5-14B-Instruct |
| Qwen/Qwen3-30B-A3B | 30B | 2025-04-29 | ✓ | Qwen3-30B-A3B |
| Qwen/Qwen2.5-32B-Instruct | 32B | 2024-09-19 | ✓ | Qwen2.5-32B-Instruct |
| Qwen/Qwen2.5-72B-Instruct | 72B | 2024-09-19 | ✓ | Qwen2.5-72B-Instruct |
| meta-llama/Llama-3.3-70B-Instruct | 70B | 2024-12-06 | ✓ | Llama-3.3-70B-Instruct |
| deepseek-ai/DeepSeek-V3 | - | 2024-12-26 | ✓ | DeepSeek-V3 |
| gpt-4o (OpenAI) | - | 2024-05-13 | × | gpt-4o/ |
| gpt-4o-mini (OpenAI) | - | 2024-07-18 | × | gpt-4o-mini |
| claude-3.7-sonnet (Anthropic) | - | 2025-02-25 | × | claude-3.7-sonnet |
| grok-3-beta (xAI) | - | 2025-02-19 | × | grok-3-beta |
| o1 (OpenAI) | - | 2024-12-05 | × | o1 |
| o3 (OpenAI) | - | 2025-04-16 | × | o3 |
| deepseek-ai/DeepSeek-R1-Distill-Qwen-7B | 7B | 2025-01-20 | ✓ | DeepSeek-R1-Distill-Qwen-7B |
| deepseek-ai/DeepSeek-R1-Distill-Qwen-32B | 32B | 2025-01-20 | ✓ | DeepSeek-R1-Distill-Qwen-32B |
| Qwen/QwQ-32B-Preview | 32B | 2025-03-06 | ✓ | QwQ-32B-Preview |
| deepseek-ai/DeepSeek-R1-Distill-Llama-70B | 70B | 2025-01-20 | ✓ | DeepSeek-R1-Distill-Llama-70B |
| deepseek-ai/DeepSeek-R1 | - | 2025-01-20 | ✓ | DeepSeek-R1 |

## D.2 MULTI-AGENT SYSTEM PERFORMANCE

We evaluate 10 multi-agent systems (MAS), including both general-purpose MAS (e.g., AgentVerse, LLM Debate) and code-specific designs (e.g., EvoMAC, MapCoder). As detailed in Table 3, we compare each MAS against a single-agent baseline on execution success (Pass@1), total API call

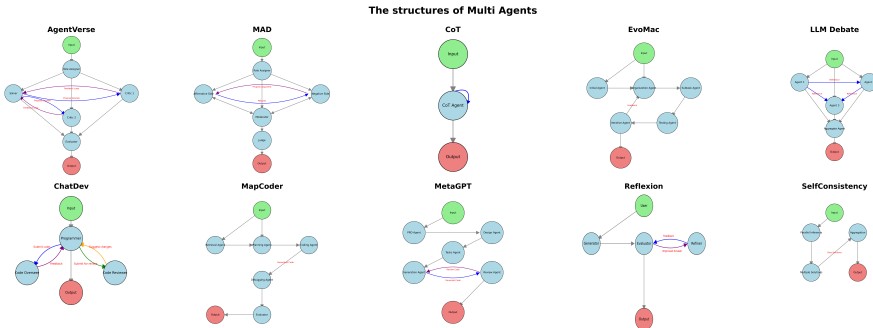

Figure 13: Multiagent system workflow visualization.

count, and cost-efficiency. Results show that while MAS can outperform single agents on complex tasks, simple strategies (e.g., Self-Refine) often strike a better balance between performance and resource usage than workflow-heavy systems like ChatDev. We visualize the MAS in the following Figure 13.

# E   ANALYSIS

## E.1   ANALYSIS OF PRD QUALITY

Our Project Requirement Descriptions (PRDs) are primarily derived from the original docstrings found within the repository source code. To enhance the quality and utility of these PRDs, we employed GPT-4o to refine and improve the original docstrings. To objectively assess this improvement, we recruited two domain experts to evaluate both the original and GPT-4o-enhanced docstrings across 100 randomly selected samples from SWE-Dev.

The experts rated each docstring on three critical dimensions—Clarity, Completeness, and Action-ability—using a 0 to 5 scale, where higher scores indicate superior quality. The human evaluation guideline is shown in Figure 27.

Participants were fully informed about the evaluation process and the nature of the task. The assessment involved only reviewing documentation and posed no ethical or privacy risks, adhering strictly to ethical standards for research involving human subjects. This evaluation provides a rigorous measure of how GPT-4o-refined docstrings enhance PRD quality in SWE-Dev.

## E.2   EXPLANATION ON THE UNDERPERFORMANCE OF REASONING MODELS

**Instruction Following Rate (IFR).** Previous experiments have shown that reasoning models perform poorly on SWE-Dev. To investigate the reasons behind this, we analyzed the instruction-following ability of these models. We measured the percentage of code files that meet the PRD requirements for each model's generated code as a metric of instruction following rate (IFR). The metric is formally defined as:

$$\text{IFR} = \frac{1}{n} \sum_{i=1}^{n} \frac{|\mathcal{G}_i \cap \mathcal{T}_i|}{|\mathcal{T}_i|}$$

where $n$ denotes the total number of tasks, $\mathcal{G}_i$ represents the set of files generated by the model for task $i$, and $\mathcal{T}_i$ denotes the set of ground truth files required by the PRD for task $i$.

To further explore this, we compared reasoning models with their chatbot counterparts by evaluating their instruction following rate. Specifically, in Figure 14, the x-axis represents the instruction following rate, and the y-axis shows the performance of both reasoning models and their chatbot counterparts on tasks where their instruction following rate is 100%.

As shown in the figure, we see that: (i) Reasoning models generally have a lower instruction-following rate compared to their chatbot version, which explains why they underperform when

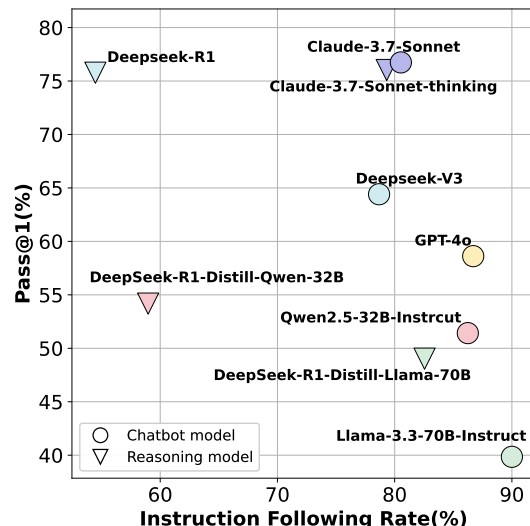

Figure 14: Comparison of reasoning and chatbot LLMs' IFR and performance on SWE-Dev. Reasoning models tend to outperform chatbots when they fully follow instructions, though their overall IFR is lower.

handling multiple tasks simultaneously. Reasoning models tend to struggle with tasks on SWE-Dev that involve performing several steps in a single call, resulting in poorer performance overall. (ii) However, on tasks where both reasoning models and their chatbot versions have an instruction-following rate of 100%, reasoning models typically outperform the chatbots. This indicates the potential of reasoning models when they can fully adhere to instructions. (iii) Claude 3.7-Sonnet is an exception to this trend, as both its reasoning and chatbot versions exhibit similar instruction-following rates and performance, which contributes to Claude's superior results.

### E.3 ERROR ANALYSIS

Figure 15 presents the distribution of failure types for both single-agent and multi-agent systems on SWE-Dev. We sample 500 samples for error analysis and categorize errors into five types: Incomplete, Logic, Syntax, Parameter, and Others, see error classification prompt in Figure 24. Across both agent types, the most prevalent error is the Incomplete Error, where models fail to implement PRD-required functions—indicating persistent challenges in task decomposition and execution coverage.

For single-agent models, Logic Errors are the second most common, followed by Parameter Errors and Syntax Errors. Interestingly, GPT-4o and Claude-3.7 show relatively fewer Syntax Errors, suggesting better adherence to Python syntax, while smaller models like GPT-4o-mini show higher incidence of both Syntax and Parameter issues, reflecting their limited reasoning capacity and weaker control over function signatures.

In contrast, multi-agent systems exhibit a different pattern. While they reduce Incomplete Errors to some extent, they often incur higher Logic or Syntax Errors—especially in methods like MAD and Self-consistency—suggesting that while agents may cover more PRD content, coordination breakdown or hallucinated reasoning steps can introduce new failure modes.

Overall, the analysis highlights the need for improved function selection, robust reasoning alignment, and stronger control over generation structure—especially in collaborative multi-agent settings.

### E.4 LIMITATION AND FUTURE WORK

**Language Scope.** SWE-Dev currently targets Python, which, while widely used, does not reflect the full diversity of real-world programming languages. A natural extension is to support other major languages such as Java, JavaScript, and C++, enabling broader evaluation and enhancing generality.

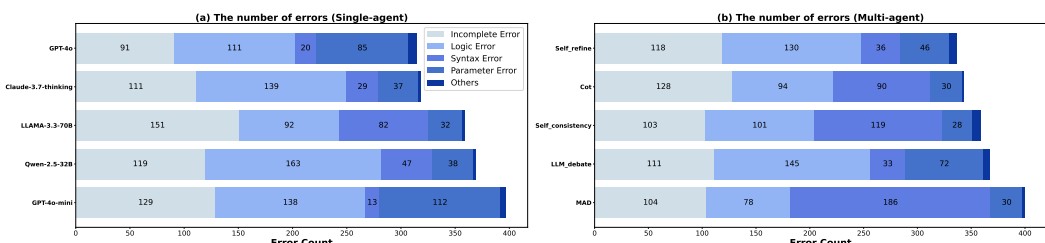

Figure 15: Failure case distribution of Single and Multi-agent.

**Training Exploration.** Our training experiments focus on standard techniques—SFT, RL, and role-wise MAS training—which yield modest gains. Future work could explore stronger RL Shao et al. (2024); Yu et al. (2025); Hu et al. (2025a), dynamic agent coordination Ye et al. (2025b), and curriculum learning Bengio et al. (2009). Notably, SWE-Dev offers fine-grained complexity signals via call trees that can guide complexity-aware training.

### E.5 BROADER IMPACTS

SWE-Dev is the first dataset tailored for autonomous feature-driven software development, addressing the gap between current automated coding and real-world software engineering demands. By providing large-scale, realistic tasks based on real repositories with executable tests, it enables rigorous and reliable evaluation of automated AI coding systems. SWE-Dev promotes the creation of more capable methods for complex software, driving innovation that can lower development costs and enhance software quality industry-wide.

## F DETAILED BENCHMARK CONSTRUCTION

### F.1 CALL TREE GENERATION

To accurately localize the implementation logic associated with each test case, we construct a call tree that captures the dynamic execution path from the test to the relevant source functions. This tree serves as the foundation for identifying the core feature logic and determining task complexity.

Figure 16 shows a generated call tree for the file `test_az.py`, which contains multiple test functions such as `test_cardinal`, `test_year`, and `test_ordinal_num`. Each test function serves as a root for its own call path, triggering downstream functions like `Num2Word_AZ.to_cardinal` and `Num2Word_AZ.int_to_word`. This tree structure reveals the multi-level and cross-functional logic activated during test execution, illustrating how test files connect to multiple feature implementations across the codebase.

We use the call tree in two key ways:

1. To select target functions for masking during task generation, enabling controllable task complexity.

2. To trace which source files and logic a model must understand to solve the task, supporting fine-grained evaluation and curriculum learning.

### F.2 DOCSTRING AUGMENTATION PROMPT

To ensure high-quality task specifications, we augment original function-level docstrings using GPT-4o. Figure 25 shows the prompt we use to generate concise, informative docstrings conditioned on the full code context.

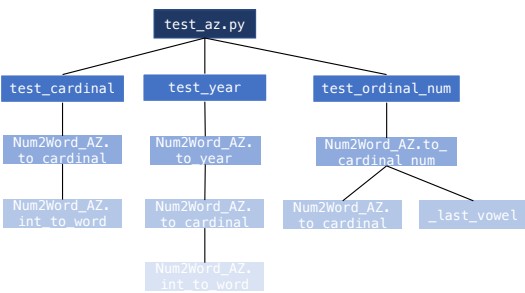

Figure 16: Example of a generated call tree for `test_az.py`.

### F.3 MASKING PROCEDURE AND REPRODUCIBILITY

Our dataset construction does not rely on any task-complexity thresholds such as call-tree depth or node count. Instead, we apply a fixed set of masking rules uniformly across all repositories. Specifically, for each task we derive the dynamic call chain exercised by the upstream test and construct a masked input using one of three modes: (i) masking depth-1 target functions, (ii) masking depth-2 dependent functions, or (iii) masking the entire call chain for call-tree depth $\geq 2$. These masking modes are predetermined and do not involve any further filtering or parameter tuning.

## G EXTENDED RELATED WORK

### G.1 MULTI-AGENT SYSTEM

For complex SE tasks that strain the context handling of single agents, Multi-Agent Systems (MAS) utilizing collaborating LLMs are an emerging research avenue. Existing frameworks like MetaGPT, ChatDev, and AgentVerse often rely on predefined agent roles and fixed interaction protocols. While effective on specific tasks, their hand-crafted structure limits generalization. Recent research explores trainable MAS, aiming for agents that dynamically adapt their organization or communication strategies. However, empirical studies of such adaptive MAS are largely constrained by benchmark complexity; evaluations are often confined to small-scale or synthetic tasks due to the lack of benchmarks providing complex interaction scenarios and reliable execution feedback. SWE-Dev's scale, complexity, and provision of executable feedback (via unit tests) establish it as the first testbed capable of supporting the training and evaluation of dynamic MAS on realistic, multi-file feature development scenarios.

## H EXPERIMENT SETTINGS

### H.1 INFERENCE

**LLMs.** We evaluate 17 chatbot LLMs with different model size, including Qwen2.5-Instruct models 1.5B/3B/7B/14B/32B/72B Qwen et al. (2025b), Qwen3 models 8B/30B-A3B Qwen (2025), Llama 3.1-8B/3.3-70B-Instruct Grattafiori et al. (2024), Phi 4 Abdin et al. (2024b), Claude-3.7-Sonnet Anthorpic (2025), Deepseek-V3 DeepSeek-AI et al. (2025), GPT-4o Hurst et al. (2024), Deepseek-Coder-V2-Lite-Instruct DeepSeek-AI et al. (2024), Qwen2.5-Coder-14B-Instruct Hui et al. (2024). Additionally, We extend the evaluation to reasoning models, including Deepseek-R1-distill models (Qwen2.5 7B/32B, Llama-70B) Guo et al. (2025), Qwen3 8B/30B-A3B (thinking) Qwen (2025), QwQ-32B-Preview, Deepseek-R1 Guo et al. (2025), OpenAI-o1 OpenAI et al. (2024), Claude-3.7-Sonnet-thinking Anthorpic (2025), and Grok-3-Beta X (2025).

**Multi-Agent Systems.** To provide a more comprehensive evaluation of SWE-Dev, we expand our study to include multi-agent systems (MAS) built on LLMs. Prior research has demonstrated that MAS can enhance performance on tasks requiring multi-step reasoning and coordination Ye et al. (2025b); Hong et al. (2024); Qian et al. (2024). In our experiments, all MAS are implemented using GPT-4o-mini OpenAI (2024) as the underlying model to ensure consistency across methods.

And for fair comparison, we utilize MASLab Ye et al. (2025a), a unified framework integrating multiple MAS implementations. We evaluate coordination-based MAS such as LLM Debate Du et al. (2023), Self Refine Madaan et al. (2023a), Multi-Agent Debate (MAD) Liang et al. (2024), and Self Consistency Wang et al. (2023a) that feature relatively simple agent interaction strategies. We further include structured, workflow-oriented MAS designed for code generation, including Agentverse Chen et al. (2023), MetaGPT Hong et al. (2024), ChatDev Qian et al. (2024), MapCoder Islam et al. (2024), and EvoMAC Hu et al. (2025b).

## H.2 TRAINING

## H.3 SINGLE-AGENT SUPERVIED FINE-TUNING

We fine-tune the model using LoRA, applying low-rank adaptations (rank $r = 16$, scaling $\alpha = 16$, $dropout = 0.05$) to the query, key, value, and output projection matrices of each attention sublayer. Training is performed with a learning rate of $6 \times 10^{-4}$ and a batch size of 32 sequences per gradient step, for up to 4 epochs. Checkpoints are saved every 50 steps, and the best model is selected based on validation loss over a held-out set of 100 examples. Fine-tuning is initialized from Qwen2.5-7B-Instruct and completed within 20 hours using 8 NVIDIA A100 GPUs. We leverage DeepSpeed Ulysses and Flash Attention to support efficient training with long input contexts.

## H.4 SINGLE-AGENT REINFORCEMENT LEARNING

For reinforcement learning (RL) training, we sampled 2k instances from SWE-Dev to balance computational feasibility and the ability to capture RL benefits. Specifically, we used Proximal Policy Optimization (PPO) Schulman et al. (2017) and Direct Preference Optimization (DPO) Rafailov et al. (2023) for training the Qwen2.5-7B-Instruct using 8 NVIDIA A100 GPUs. All RL experiments are run on a single node with 8×A100 GPUs, using VERL PPO with a vLLM-based rollout backend.

**PPO**: The training was conducted using a batch size of 256 and trained for 5 epochs, with a learning rate of $1 \times 10^{-6}$ for the actor and $1 \times 10^{-5}$ for the critic. We set the KL coefficient to 0.001. The training was set to save checkpoints every 10 steps. We used a maximum prompt length of 8192 tokens and set a micro-batch size of 32. The maximum response length is 2048 tokens. The reward for PPO is calculated based on the pass rate of the test cases. Concretely, for each rollout solution, the reward is defined as the pass rate, i.e., the number of test cases successfully passed by the solution divided by the total number of test cases available for that task. During RL training, we do not employ any external agent framework: the language model directly receives the problem requirement description (PRD) together with the relevant code context as input and generates a single rollout solution.

**DPO**: For DPO training, we applied LoRA with a rank of 64, scaling factor $\alpha = 128$, and dropout set to 0. The preference loss function (`pref_loss`) was set to sigmoid, which is commonly used in DPO for preference-based optimization. Training was performed for 5 epochs, using a batch size of 8 and a learning rate of $1 \times 10^{-5}$.

For a fair comparison with SFT in §4.2.2, we used the same 2k training samples for both SFT and RL. The details for SFT training are outlined in Appendix H.3.

These methods allow us to assess the impact of RL on model performance using the SWE-Dev dataset while maintaining efficient training..

## H.5 MINGLE-AGENT SUPERVISED FINE-TUNING

In our multi-agent fine-tuning experiments, we utilize a simplified version of EvoMAC Hu et al. (2025b), retaining only two core roles: **Organizer** and **Coder**, see Figure 20. The fine-tuning process follows an iterative workflow. Initially, the **Organizer** processes the Project Requirement Description (PRD) and breaks it down into clearly defined subtasks or instructions. Subsequently, the **Coder** generates corresponding code implementations for these subtasks. The generated code is then evaluated using the provided ground truth (GT) test cases. Feedback from these evaluations informs subsequent iterations, enabling iterative refinement of both the task decomposition by the Organizer

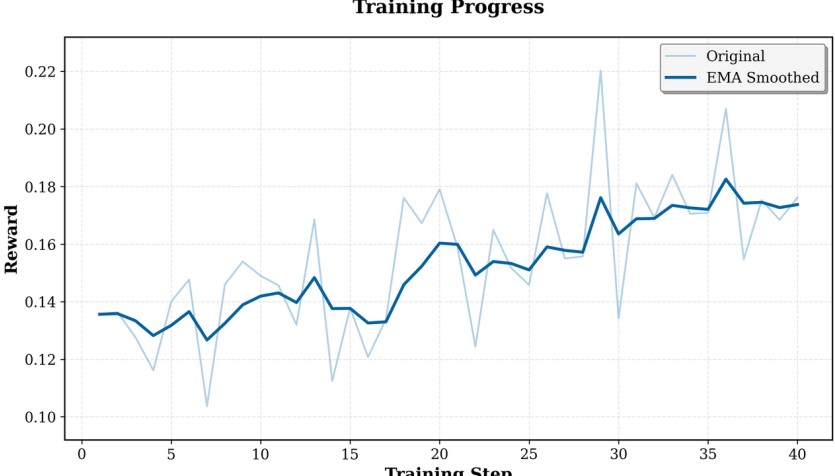

Figure 17: The PPO training progress of Qwen2.5-7B-Instrut on the SWE-Dev with 40 steps and 2048 max token length. The light curve shows the raw reward, while the dark curve applies EMA smoothing.

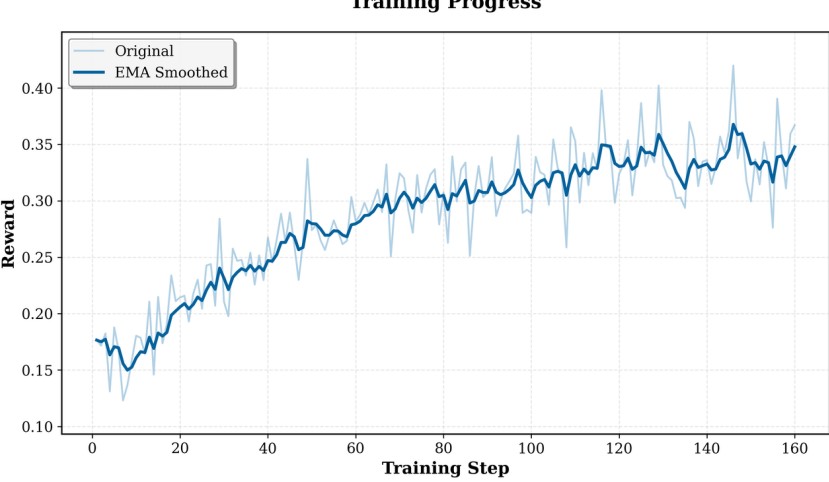

Figure 18: The PPO training progress of Qwen2.5-7B-Instrut on the SWE-Dev with 160 steps and 8192 max token length. The light curve shows the raw reward, while the dark curve applies EMA smoothing.

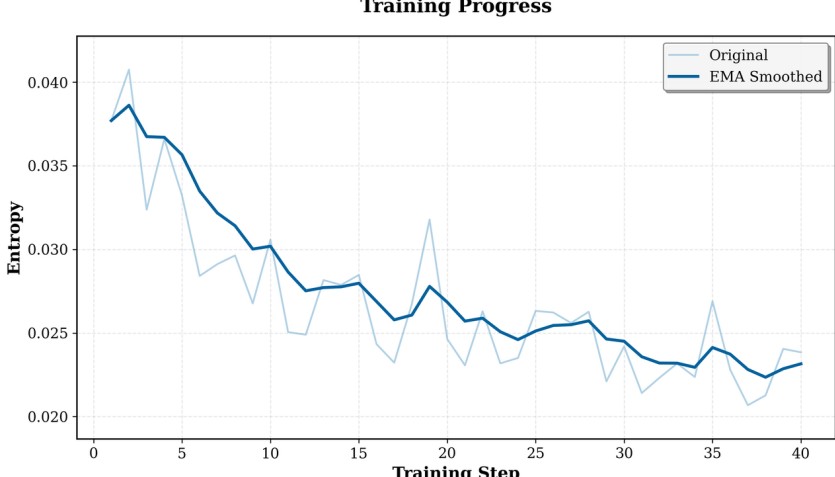

Figure 19: Entropy trajectory during PPO training (40 updates). The light curve shows the raw policy entropy, and the dark curve shows the EMA-smoothed trend. Entropy decreases steadily over training, indicating reduced exploration and a progressively sharper output distribution—consistent with the observed contraction in diversity under RL.

and the code generation by the Coder. Fig. 22 and Fig. 23 respectively show the prompts used for the Coder and Organizer.

**Rejection Sampling Procedure.**

To effectively leverage the feedback from GT test cases, we employ rejection sampling—a method widely adopted in reinforcement learning and language model fine-tuning Zhou et al. (2024); Snell et al. (2023). The detailed procedure is as follows:

1. **Iterative Reasoning with EvoMAC**: For each training instance, EvoMAC executes multiple rounds of reasoning. In each iteration, the generated code from the Coder is tested against the GT test cases to compute its performance.

2. **Selection of High-quality Trajectories**: Trajectories that show improved performance over previous iterations (as indicated by an increased pass rate on GT test cases) are selectively retained. Conversely, trajectories that do not demonstrate progress or degrade in performance are discarded. This ensures that only beneficial and constructive data is used for fine-tuning.

3. **Role-wise Fine-tuning**: The retained high-quality trajectories are utilized to separately fine-tune the Organizer and the Coder. Specifically, the Organizer is trained to better structure and decompose tasks from PRDs, while the Coder is refined to enhance code generation capabilities for defined subtasks. This role-specific fine-tuning promotes specialization and improves overall performance.

As shown in §4.2.3, through this simplified EvoMAC and structured rejection sampling approach, our multi-agent fine-tuning effectively enhances the capabilities of each agent, contributing to significant performance gains on SWE-Dev.

## I  LICENSING

All codebases and data used in this work are sourced from publicly available GitHub repositories. We have ensured compliance with the corresponding licenses of these repositories, respecting all terms of use and attribution requirements.

## Multi-agent Fine-tuning

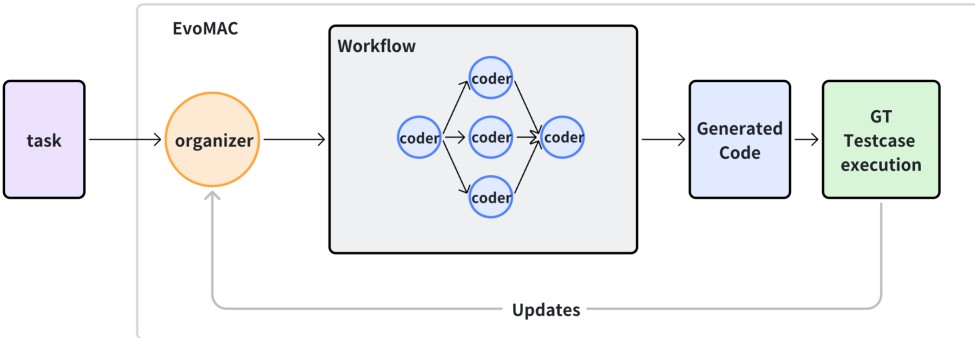

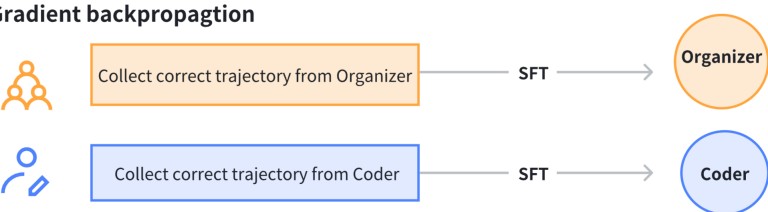

Figure 20: Overview of Multi-agent Fine-tuning in EvoMAC. This framework leverages both textual and gradient-based supervision to improve multi-agent collaboration. During inference, an organizer assigns roles and coordinates a team of coders to generate code, which is then validated using ground truth test cases. Successful execution trajectories are collected and used to fine-tune both the organizer and coders individually via supervised learning, enabling role-specific optimization for complex software development tasks.

---

**Single LLM inference prompt**

```
# AIM: You need to assist me with a Python package feature
    development task. I will provide a Product Requirements Document
     (PRD) that details the functionality and lists the empty
    functions that need to be implemented across different file
    paths. I will also provide the complete "Code Context" of all
    files mentioned in the PRD.

Your task is to implement ONLY the empty functions described in the
     PRD while preserving ALL OTHER CODE in the provided files
    exactly as is. This is absolutely critical - you must keep all
    imports, class definitions, functions, comments, and other code
    that is not explicitly mentioned in the PRD for implementation.

When implementing the functions:
1. Carefully identify which functions from the PRD need
    implementation.Implement them based on the docstrings and
    specifications in the PRD
2. Do not add any new "import" statements unless absolutely
    necessary
3. Do not modify ANY existing code structure, only implement the
    empty functions

For each file mentioned in the PRD, you MUST output the COMPLETE
    file code with your implementations inserted. Your output format
     must follow this exact pattern:

# OUTPUT FORMAT FOR EACH FILE:
@ [relative path/filename]
```python
[COMPLETE file code including ALL original code plus your
    implementations]
```
@ [relative path/filename]
```python
[COMPLETE file code including ALL original code plus your
    implementations]
```

IMPORTANT: Make sure your output includes EVERY function, class,
    import statement, and comment from the original code context.
    The only difference should be that the empty functions specified
     in the PRD are now implemented.

# PRD:
{PRD}

# Code Context:
{code_snippet}
```

Figure 21: Single LLM Inference Prompt.

## J PROMPTS

This section includes all prompts used in the generation, evaluation and analysis process.

---

**EvoMAC Coding Agent Prompt**

```
You are Programmer. we are both working at ChatDev. We share a
    common interest in collaborating to successfully complete a task
     assigned by a new customer.

You can write/create computer software or applications by providing
     a specific programming language to the computer. You have
    extensive computing and coding experience in many varieties of
    programming languages and platforms, such as Python, Java, C, C
    ++, HTML, CSS, JavaScript, XML, SQL, PHP, etc,.

Here is a new customer's task: {task}.

To complete the task, you must write a response that appropriately
    solves the requested instruction based on your expertise and
    customer's needs.

According to the new user's task and you should concentrate on
    accomplishing the following subtask and pay no heed to any other
     requirements within the task.

Subtask:
{subtask}.

Programming Language: python,

Codes:
{codes}
```

Figure 22: Role Prompt for EvoMAC Coding Agent.

**EvoMAC Organizing Agent Prompt**

```
As the Leader of a coding team, you should analyze and break down
    the problem into several specific subtasks and assign a clear
    goal to each subtask. Ensure each subtask is extremely detailed,
     with a clear description and goal, including the corresponding
    PRD statement where necessary.

The workflow should be divided into minimal, executable subtasks,
    each involving one method implementation. The target_code should
     only contain the relative paths and function names for the
    specific code that is required for that subtask.

Each subtask should be assigned a unique task_id, and the
    description should reflect the exact requirements of the PRD
    corresponding to that method or task. The target_code should be
    precise, containing only the specific Python code (relative path
     and method/function name) that corresponds to the subtask's
    scope.

The format should strictly follow the JSON structure below:

```json
[
    {
        "task_id":"1",
        "description":"Task Description",
        "target_code": [
            "relative_python_path:function_name",
            "relative_python_path:class_name.method_name"
        ]
    }
]
```

Use the backticks for your workflow only.

Note:

(1) Each subtask should be self-contained and represent one method'
    s implementation.

(2) The `description` should be based on specific statements from
    the PRD, and it must explain what the subtask is aiming to
    achieve.

(3) The `target_code` should only reference the code paths and
    function names for the methods to be implemented for the subtask
    .

(4) The number of subtasks should not exceed 5. Some tasks might
    combine multiple smaller functions if needed to fit within the
    limit.

(5) Each subtask is handled independently by different agents, so
    the description should be thorough, ensuring clarity even
    without the full context of the PRD.
```

Figure 23: Role Prompt for EvoMAC Organizing Agent.

**Error Classification Prompt**

```
You are an error classification expert. Based on the provided PRD,
    LLM-generated code, and error message, your task is to analyze
    and categorize the primary issue.

1. Analyze the root cause of the problem using the PRD, the code,
    and the error message.
2. If multiple issues exist, return only the most severe and
    primary one (return exactly one ProblemType).
3. Return the result in strict JSON format with the following
    structure:
{
    "ProblemType": {
        "MainCategory": "Main error category",
        "SubCategory": "Specific sub-category of the issue",
        "Reasoning": {
            "SymptomAnalysis": "Observed abnormal behavior (in Chinese)
                ",
            "RootCause": "Attribution analysis combining PRD and code
                (in Chinese)",
            "ErrorMechanism": "Technical explanation of how the error
                occurs (in Chinese)"
        }
    }
}

Below is the data provided to you:
PRD:
{prd}

Generated Code:
{results}

Error Message:
{input_text}

Please ensure your response strictly follows the JSON format above.

The allowed values for MainCategory are limited to the following
    five options - read them carefully and choose the most
    appropriate one:
1. Logic Error: Logical errors such as assertion failures or
    failure to meet PRD requirements.
2. Syntax Error: Syntax issues such as unexpected tokens,
    indentation problems, etc.
3. Parameter Error: The function required by the PRD is present,
    but input/output parameters are incorrect or missing.
4. Incomplete Error: Some required functions are entirely missing
    as per the PRD. Make sure to distinguish between a truly missing
     function and one that exists but contains logic or syntax
    errors.
5. Others: Any other issues that do not fit the above categories.

You must carefully select the MainCategory to ensure accuracy.
Do not return any MainCategory that is not listed above.
Do not return an empty MainCategory.
```

Figure 24: Prompt Template for Error Type Classification.

```
Docstring Augmentation Prompt

# Context: The following Python code is provided for reference. It
    includes functions, classes, and other elements that provide
    context for the function or class below. Additionally, any
    constants or variables defined outside functions/classes are
    considered as part of the context and should be explained if
    used.

# Full Code:
'''python
{full_code}
'''

# Code for {name}:
'''python
{code_snippet}
'''

# Docstring:
Please generate a concise and clear docstring for the above {name}
    based on the full code context. Ensure the docstring briefly
    explains the {name}'s purpose, parameters, return values, and
    any relevant dependencies or interactions with other parts of
    the code. If there are any constants or variables used within
    the {name}, explain their role and significance, including where
     they are defined and how they interact with the function or
    class.

For functions: describe the input parameters, expected output, and
    any important side effects in a few sentences. Also, explain any
     constants used inside the function (if applicable).

For class.methods: describe the input parameters, expected output,
    and any important side effects in a few sentences. Also, explain
     any constants used inside the function (if applicable).

For classes: describe the main attributes and methods, along with
    the general purpose of the class in a brief summary. Mention any
     constants used in the class and explain their purpose and how
    they interact with class methods and attributes. Keep the
    docstring focused, avoiding unnecessary details or repetition.

# Output format
Your response should strictly follow the format below, without any
    other text or comments.
\"\"\"
docstring
\"\"\"
```

Figure 25: Docstring Augmentation Prompt in Task Generation.

## Categories for Classifying Packages

```
You are a Python expert. Given the name of a PyPI package, classify
    it into ONE category from the list below based on its MOST
    central and primary purpose.

Categories:

1. Web & Network Automation
Packages that support automation of web browsing, API communication,
    and network protocols.
Criteria: Enables browser control, HTTP requests, network
    operations, or web server handling.

2. Data Processing & Integration
Packages that extract, parse, or convert structured/unstructured
    data formats.
Criteria: Handles parsing or converting text, JSON, YAML, XML, or
    dates.

3. Security & Access Control
Packages that focus on authentication, authorization, or access
    control mechanisms.
Criteria: Implements rules, policies, or authentication methods.

4. Command-Line & Developer Tools
Packages that assist in building CLI tools, test frameworks, or
    code quality analysis.
Criteria: Aimed at improving the development experience, command-
    line interfaces, or code quality.

5. Cloud & Data Storage
Packages interacting with cloud services, databases, or data
    storage solutions.
Criteria: Provides interfaces or tools to access, manage, or
    validate remote data or cloud resources.

6. Data Science & Visualization
Packages used for scientific computing, visualization, or
    statistical evaluation.
Criteria: Supports data analysis, visualization, or scientific
    research.

7. Others
Packages that do not clearly belong in the other categories or are
    too general/specialized.
Criteria: Doesn't strongly align with the definitions above or
    serves a unique/niche purpose.

Please output only the category number (only one category), no
    explanation unless asked. Choose the single best fit.

Package name: {package_name}

You must strictly follow the format below, only a number no other
    text:
1
```

Figure 26: Prompt Template for Classifying Packages.

---

**Human Evaluation Guideline for PRD Quality**

```
Each docstring is evaluated independently along the following three
    dimensions, using a 0-5 scale (0 = very poor, 5 = excellent):

Clarity - How easy the docstring is to understand for a competent
    software engineer. Consider language clarity, readability, and
    absence of ambiguity.
Completeness - Whether the docstring provides all necessary
    information to understand the function's behavior. Consider
    whether inputs, outputs, parameters, and important logic are
    described.
Actionability - How effectively the docstring guides actual
    implementation. Consider whether a developer could use the
    docstring alone to reasonably implement the function.

Rating Scale:
5: Excellent - No issues; highly clear, complete, and actionable.
4: Good - Minor improvements possible.
3: Fair - Understandable but lacking in one area.
2: Poor - Vague or missing key information.
1: Very Poor - Hard to follow or largely unhelpful.
0: Unusable - Cannot inform implementation at all.

If the original docstring is missing or boilerplate-only, please
    rate accordingly. Docstrings are to be rated individually
    without direct comparison.
```

Figure 27: Human Evaluation Guideline for PRD Quality.