# OpenReview forum: "SWE-Dev: Training and Evaluating Autonomous End-to-End Feature-Driven Software Development"
_ICLR.cc/2026/Conference — ICLR 2026 Conference Desk Rejected Submission_

### Official Review · Reviewer_icEc · 2025-10-28

**Soundness:** 3
**Presentation:** 2
**Contribution:** 3
**Rating:** 4
**Confidence:** 4

**Summary:**

The paper introduces SWE-Dev, a large repository-level dataset for end-to-end feature-driven software development. It contains 14,000 training and 500 test tasks, each with a runnable environment and developer-authored unit tests, enabling execution-based evaluation and training. Experiments span 17 chatbot LLMs, 10 reasoning models, and 10 multi-agent systems. The authors report that strong closed models still struggle (e.g., Claude-3.7-Sonnet ~24.25% Pass@3 on Hard) and that SFT/RL yield modest gains; they also show simple MAS often outperform heavier agent frameworks.

From this reviewer's perspective, the dataset collection approach is compelling and addresses a timely need in the community. However, the evaluation protocol, training setups, and subsequent conclusions raise concerns about the validity of the design choices and, thus, their generalizability. Lack of comparison with recent datasets and benchmarks raises a question of novelty.

**Strengths:**

1. Examples of high-quality, diverse SWE tasks are still limited. SWE-Dev makes a valuable contribution to this domain by providing over 14,000 tasks with a specific focus on feature development.
2. While GitHub issues can include feature requests, most are bug reports. The proposed methodology is well-suited for generating feature-development tasks, which helps correct the existing bias towards bug-fixing in other datasets.
3.  The call tree serves as a compelling method for differentiating task complexity. Figure 8a demonstrates that this approach is indeed effective for adjusting task difficulty.

**Weaknesses:**

1. The related work section on LLMs for coding does not fully cover recent, relevant works. The authors claim that SFT and RL have "largely focused on function-level tasks," but this overlooks several works [1, 2, 3] that apply RL at scale to complex SWE tasks.
2. The paper lacks a comparison with [4], a popular large-scale dataset with SWE tasks for training, especially given its similar approach of using tests and AST for task generation.
3. The paper claims key distinguishing characteristics ("Realistic scale," "Robust evaluation," "Verifiable training set") but fails to compare against [5], where similar claims are made.
4. Adding [6] to the benchmark comparison in Table 1 would provide greater transparency.
5. The paper does not appear to address common failure modes of synthetic dataset generation. For example, it is unclear if the test suites sometimes imply a specific class or function name that an agent could only guess, rather than deduce.
6. The LLM evaluation methodology is simplistic and not representative of current best practices. SOTA evaluation typically involves running multi-step agents (e.g., SWE-agent, OpenHands). Even simpler agentic approaches like Agentless involve separate steps for localization and generation. The conclusions drawn from the paper's evaluation and subsequent training raise concerns about their generalizability.
7. The described methodology is not language-agnostic, as it relies on Python-specific methods for code tracing. This limits its generalizability to other programming languages.
8. Some evaluation results seem contradictory, such as Qwen2.5-72B-Instruct performing similarly to Claude-3.7-Sonnet on the easy subset.
9. The RL training parameters are questionable. If the 2k-task dataset is used for 5 epochs, this yields 10k total trajectories. With a batch size of 256, this results in only ~40 training steps, which seems insufficient to achieve reliable improvements.

[1] Michael Luo et al. DeepSWE: Training a Fully Open-sourced, State-of-the-Art Coding Agent by Scaling RL.

[2] Alexander Golubev et al. Training Long-Context, Multi-Turn Software Engineering Agents with Reinforcement Learning.

[3] Shiyi Cao et al. SkyRL-v0: Train Real-World Long-Horizon Agents via Reinforcement Learning.

[4] John Yang et al. SWE-smith: Scaling Data for Software Engineering Agents.

[5] Ibragim Badertdinov et al. SWE-rebench: An Automated Pipeline for Task Collection and Decontaminated Evaluation of Software Engineering Agents.

[6] Naman Jain et al. R2E-Gym: Procedural Environment Generation and Hybrid Verifiers for Scaling Open-Weights SWE Agents.

**Questions:**

1. Could the authors elaborate on the process described as "We use structural properties of the call tree (e.g., depth and node count) to identify key function nodes"? Specifically, how do you differentiate nodes that represent core feature logic from common utilities or helper functions?
2. Can you provide a more detailed analysis of why the evaluated reasoning models have a lower Instruction Following Rate (IFR)? What are the authors' insights on this observed behavior?

---

> ### Author Response · Authors · 2025-11-26
> **[Part 1/3] Author Response**
>
> We sincerely appreciate your time and thoughtful comments. Below, we respond to each point in detail.
>
> ---
>
> **[Weakness 1]**
>
> The authors claim that SFT and RL have "largely focused on function-level tasks," but this overlooks several works that apply RL at scale to complex SWE tasks.
>
> **Answer**:
>
> We thank the reviewer for highlighting these relevant RL-based SWE works; we have added and discussed them in the Appendix B.
>
> ---
>
> **[Weakness 2 -- Weakness 4]**
>
> The paper lacks a comparison with other related papers like SWE-Smith, R2E-Gym and SWE-rebench:
> - The paper lacks a comparison with [1], a popular large-scale dataset with SWE tasks for training, especially given its similar approach of using tests and AST for task generation.
> - The paper claims key distinguishing characteristics ("Realistic scale," "Robust evaluation," "Verifiable training set") but fails to compare against [2], where similar claims are made.
> - Adding [3] to the benchmark comparison in Table 1 would provide greater transparency.
>
> **Answer**:
>
> We thank the reviewer for pointing out the need for a more comprehensive comparison. The table below shows a comparison with other datasets. SWE-Dev is only the dataset targeted on feature development with most range of lines of code to modify. Below we summarize the key differences.
>
> |                | **Task type**              | **Lines of Modified Code** | **Real-world Test Files** | **# Repo** | **# Tasks** |
> |----------------|------------------------|------------------------|-----------------|--------------|-------|
> | **SWE-Smith**[1]   | Bug Fix                | 9.58                   | ✅               | 28           | 50k   |
> | **R2E-Gym**[2]     | Issue solving          | 5.72                      | ❌               | 13           | 8k    |
> | **SWE-rebench**[3] | Issue solving          | 37                     | ✅               | 3468         | 21k   |
> | **SWE-Perf**[4]    | Performance Efficiency | 131.1                  | -               | 9            | 140   |
> | **PerfBench**[5]   | Performance Efficiency | 103.7                  | -               | 32           | 81    |
> | **SWE-Dev**        | Feature development    | 190.0                  | ✅               | 1061         | 13k   |
>
> 1. **Task setting.**
>      As shown in the new comparison table, these datasets target task types that are different from SWE-Dev:
>     - SWE-Smith, R2E-Gym, SWE-ReBench, and SWE-bench-style benchmarks focus on **issue solving / bug fixing**, which usually involves limited code to modify. SWE-Perf and PerfBench target **performance optimization** tasks.
>     - In contrast, SWE-Dev is designed for **feature implementation**: given a natural-language PRD, the model must add new functionality, often modifying ~190 lines of code across multiple files in a real repository.
> 2. **Construction methodology and evaluation scale.** Their construction methodologies and evaluation scales also differ substantially.
>     - SWE-Smith relies on static AST-based transformations to break existing tests and generate many training instances per repository.
>     - R2E-Gym procedurally generates executable environments and synthetic tests from commits, and SWE-ReBench continuously mines GitHub issues/PRs into interactive tasks.
>     - SWE-Perf and PerfBench curate performance-related PRs and build custom harnesses for performance evaluation, but they limit in a small scale.
>     - In contrast, SWE-Dev uses dynamic execution tracing to locate feature related code on upstream human-written tests and generate tasks by masking them. This yields feature-level, multi-file tasks that are guaranteed to be solvable in a real environment, while still scaling to 13k tasks over 1,061 repositories with real test suites.
>
>
> ---
>
> **[Weakness 6]**
>
> The LLM evaluation methodology is simplistic and not representative of current best practices. SOTA evaluation typically involves running multi-step agents (e.g., SWE-agent, OpenHands). Even simpler agentic approaches like Agentless involve separate steps for localization and generation. The conclusions drawn from the paper's evaluation and subsequent training raise concerns about their generalizability.
>
> **Answer**:
>
> We thank the reviewer for this concern. To address it, we evaluated mainstream agent frameworks commonly used on SWE-bench, including Agentless and OpenHands, on the SWE-Dev test set. All experiments used gpt-oss-120b with reasoning_effort = high and a 32,768 maximum output tokens. For OpenHands, we additionally capped the agent to 100 reasoning steps. Each setting was run three times, and the table shows the averaged results.
>
> |               | **Easy Pass@1** | **Hard Pass@1** |
> |---------------|:--------:|:--------:|
> | **Agentless** |  63.14%  |  34.84%  |
> | **OpenHands** |  57.95%  |  31.97%  |

---

> ### Author Response · Authors · 2025-11-26
> **[Part 2/3] Author Response**
>
> **[Weakness 7]**
>
> The described methodology is not language-agnostic, as it relies on Python-specific methods for code tracing. This limits its generalizability to other programming languages.
>
> **Answer**:
>
> We thank the reviewer mentioning this important feature: language-agnostic. We agree that our implementation is currently Python-based, but the methodology itself is not tied to Python, we will explain this in the following two aspects:
>
> 1. **While our implementation uses Python’s trace module, the methodology itself is not Python-specific.**
>     It only requires two generic capabilities: runtime instrumentation and function-level tracing during test execution, which exist in many languages.
> 2. **Other major languages provide equivalent mechanisms for dynamic call tracing.**
>     For example, Go offers runtime/trace and pprof, JVM languages (Java/Scala/Kotlin) provide JVMTI and bytecode instrumentation, JavaScript/TypeScript have V8 Inspector Protocol, Ruby has TracePoint, and C/C++ support call tracing via ptrace, LD_PRELOAD interception, or LLVM sanitizers.
>
>
> ---
> **[Weakness 8]**
>
> Some evaluation results seem contradictory, such as Qwen2.5-72B-Instruct performing similarly to Claude-3.7-Sonnet on the easy subset.
>
> **Answer**:
>
> We thank the reviewer for the observation. We believe the results are not contradictory, and the behavior is expected for benchmarks with varying difficulty levels.
>
> 1. **The similar performance on easy tasks is expected, not contradictory.** Easy tasks in SWE-Dev have shallow call trees and low structural complexity (Sec. 5, Fig. 8a), so many models naturally cluster at similar performance levels. HumanEval[6] is a canonical example: it is universally regarded as an “easy” benchmark, and strong models such as Qwen2.5-72B-Instruct achieving performance close to Claude-3.5/3.7-Sonnet is considered normal, not surprising.
> 2. **The separation becomes clear on hard tasks.** On hard tasks, where multi-file integration and deeper reasoning are required, Claude-3.7-Sonnet clearly outperforms Qwen2.5-72B-Instruct.
> 3. **This phenomenon is consistent with other coding benchmarks:** for example, in BigCodeBench[7], Qwen2-72B-Instruct performs comparably to Claude-3.5-Sonnet on easier problems
>
> ---
> **[Weakness 9]**
>
> The RL training parameters are questionable. If the 2k-task dataset is used for 5 epochs, this yields 10k total trajectories. With a batch size of 256, this results in only ~40 training steps, which seems insufficient to achieve reliable improvements.
>
> **Answer**:
>
> We thank the reviewer for raising this concern about the RL training configuration. We have clarified the training procedure and added the corresponding training curves in Figures 17 and Figure 18. Our findings are as follows:
> 1. **Initial 40-step run (Figure 17)**
>   In our initial experiments, due to constrained by limited resources, so we deliberately used a short RL schedule as a pilot run. As the reviewer correctly inferred, using a 2k-task dataset for 5 epochs with a batch size of 256 corresponds to roughly 40 training updates. Even under these restricted conditions, we already observed clear performance improvements from RL, suggesting that the SWE-Dev training set is sufficiently rich to support RL optimization.
> 2.  **Extended 160-step run (Figure 18)**
>   In the revised version, we further extend the RL training to 160 updates, training on H200 GPUs and increasing the output length limit to 8192 tokens. The resulting learning curve is shown in Figure 18, where the performance improvements are noticeably larger and more stable. These results further confirm the effectiveness and reliability of our RL training setup.

---

> ### Author Response · Authors · 2025-11-26
> **[Part 3/3] Author Response**
>
> **[Question 1]**
>
> Could the authors elaborate on the process described as "We use structural properties of the call tree (e.g., depth and node count) to identify key function nodes"? Specifically, how do you differentiate nodes that represent core feature logic from common utilities or helper functions?
>
> **Answer**:
>
> We apologize for the imprecise wording. We do not perform any semantic classification of functions into "core logic" vs. "helpers", nor do we use depth/node count as a semantic decision rule.
>
> 1. **What we actually do:** We select mask targets only from functions that (i) are executed by the developer-written tests (obtained via dynamic tracing) and (ii) reside in non-test source files—thus every target is causally involved in the tested behavior. Since all nodes in call tree come directly from dynamic execution tracing, which captures the exact runtime call path exercised by the upstream tests, ensuring every node is causally involved in the tested behavior.
> 2. **Depth and node count are only statistical signals we use to describe structural patterns in the analysis.** While deeper nodes tend to be utilities and shallower nodes tend to feature logic, this is merely an empirical observation rather than a rule. As illustrated in Figure 16, these patterns help interpret complexity but do not influence task construction.
>
> ---
> **[Question 2]**
>
> Can you provide a more detailed analysis of why the evaluated reasoning models have a lower Instruction Following Rate (IFR)? What are the authors' insights on this observed behavior?
>
> **Answer**:
>
> Our analysis in Appendix E.2 shows the detailed analysis
> 1. **IFR definition and observation.** We introduce the Instruction Following Rate (IFR), which measures the proportion the model generates all files required by the PRD during inference. Our observations are two folds:
>   - Reasoning models exhibit significantly lower IFR than their chat counterparts, especially on multi-file tasks. For instance, Qwen2.5-32B-Instrcut Deepseek-r1-distill Qwen2.5-32B have 58.97% and 86.24% IFR respectively while have pass rate 24.25% and 43.64% on Easy test split.
>   - When we condition on IFR = 100%, reasoning models consistently outperform their chat counterparts.
> 2. **Explanation**. We hypothesize that the lower IFR mainly comes from the difficulty of multi-file generation for reasoning models. In math-style reasoning, the model only needs a single chain of thought to solve one problem, whereas on SWE-Dev it must generate many files and repeatedly switch from one sub-task to another as it starts each new file. These topic shifts make it easy for long reasoning traces to drift away from the original PRD, so reasoning models often omit some required files, leading to a lower IFR. This is consistent with prior findings that chain-of-thought can cause reasoning drift and reduce structural completeness in code [8,9].
>
> [1] Yang, John, et al. "Swe-smith: Scaling data for software engineering agents." arXiv preprint arXiv:2504.21798 (2025).
>
> [2] Jain, Naman, et al. "R2e-gym: Procedural environments and hybrid verifiers for scaling open-weights swe agents." arXiv preprint arXiv:2504.07164 (2025).
>
> [3] Badertdinov, Ibragim, et al. "SWE-rebench: An Automated Pipeline for Task Collection and Decontaminated Evaluation of Software Engineering Agents." arXiv preprint arXiv:2505.20411 (2025).
>
> [4] He, Xinyi, et al. "Swe-perf: Can language models optimize code performance on real-world repositories?." arXiv preprint arXiv:2507.12415 (2025).
>
> [5] Garg, Spandan, Roshanak Zilouchian Moghaddam, and Neel Sundaresan. "PerfBench: Can Agents Resolve Real-World Performance Bugs?." arXiv preprint arXiv:2509.24091 (2025).
>
> [6] Chen, Mark. "Evaluating large language models trained on code." arXiv preprint arXiv:2107.03374 (2021).
>
> [7] Zhuo, T. Y., et al.(2025). BigCodeBench: Benchmarking Code Generation with Diverse Function Calls and Complex Instructions. In Proceedings of the International Conference on Learning Representations (ICLR 2025).
>
> [8] Fu, Tingchen, et al. "Scaling reasoning, losing control: Evaluating instruction following in large reasoning models." arXiv preprint arXiv:2505.14810 (2025).
>
> [9] Zhu, Yuqi, et al. "Uncertainty-Guided Chain-of-Thought for Code Generation with LLMs." CoRR (2025).
>
> **Finally:** We really appreciate your insightful suggestions, which have already helped us strengthen both the analysis and presentation of this work. We hope the above clarifications resolve the concerns. If the response is helpful, we would be grateful if you could consider adjusting your score accordingly.

---

### Official Review · Reviewer_pjSU · 2025-10-31

**Soundness:** 2
**Presentation:** 3
**Contribution:** 3
**Rating:** 6
**Confidence:** 3

**Summary:**

This paper presents SWE-Dev, a large-scale repository-level dataset designed for feature-driven software development, a realistic yet underexplored setting distinct from bug fixing or function completion.  Each task in SWE-Dev is constructed from real open-source repositories by anchoring on executable unit tests. The authors dynamically trace test execution to build function-level call trees, identify and mask the core implementation regions, and generate refined Project Requirement Descriptions (PRDs) describing the intended functionality.  The dataset comprises 14k training and 500 test samples drawn from over 1,000 repositories, each with runnable environments, enabling verifiable execution-based evaluation (Pass@k).  SWE-Dev also supports supervised fine-tuning (SFT), reinforcement learning (RL), and multi-agent (MAS) training paradigms. Experiments cover 17 chatbot LLMs, 10 reasoning-oriented LLMs, and 10 multi-agent systems, analyzing how performance varies with data scale, reasoning strategy, and task complexity (measured by call-tree depth and node count). SWE-Dev provides a realistic, reproducible, and scalable benchmark for studying end-to-end autonomous software development.

**Strengths:**

- The work targets feature addition and integration,a dominant portion of real-world programming that existing datasets fail to capture,offering a meaningful shift from isolated code completion benchmarks.

- The construction process forms a coherent, automatable pipeline with controllable task difficulty through structural parameters.

- By grounding evaluation in developer-written unit tests, the benchmark yields clear functional correctness signals rather than proxy metrics like BLEU or CodeBLEU.

- The study spans multiple LLM families and training paradigms (SFT, RL, MAS), providing cross-model trends and demonstrating SWE-Dev’s discriminative power in evaluating realistic code-generation capability.

- The dataset aligns well with real development workflows, enabling both evaluation and fine-tuning research; the design choices are transparent, and the implementation appears reproducible.

**Weaknesses:**

- The claim that “reasoning models do not always outperform standard ones” lacks supporting analysis on temperature, sampling strategy, or reflection iterations, making causality unclear.

- Although RL reportedly improves Pass@1 but reduces diversity, quantitative measures such as output entropy, candidate uniqueness, or AST variance are absent.

- The paper defines task complexity via call-tree depth and node count but does not publish explicit thresholds or masking ratios, hindering full reproducibility.

- While MAS systems are benchmarked, there is little discussion of communication structure, coordination overhead, or emergent behaviors that might explain observed differences.

**Questions:**

1. How sensitive are results to inference hyperparameters such as temperature, top-p/k, or reflection rounds?
2. Can the authors quantify the diversity trade-off in RL models using metrics like output entropy or unique-pass ratios?
3. What exact thresholds distinguish easy vs hard tasks in call-tree depth and node count, and are these thresholds consistent across repositories?
4. What evidence supports the claim that general multi-agent systems outperform code-specific ones,is this due to communication efficiency or broader role specialization?

---

> ### Author Response · Authors · 2025-11-26
> **[Part 1/3] Author Response**
>
> We sincerely appreciate your time and thoughtful comments. Below, we respond to each point in detail.
>
> ---
>
> **[Weakness 1]**
>
> The claim that “reasoning models do not always outperform standard ones” lacks supporting analysis on temperature, sampling strategy, or reflection iterations, making causality unclear.
>
>
> **Answer**:
>
> We thank the reviewer for raising this concern. Our analysis in Appendix E.2 shows that the primary cause is not decoding hyperparameters but instruction-following failure.
>
> 1. **IFR definition and observation.** We introduce the Instruction Following Rate (IFR), which measures the proportion the model generates all files required by the PRD during inference. Our observations are two folds:
>    - Reasoning models exhibit significantly lower IFR than their chat counterparts, especially on multi-file tasks. For instance, Qwen2.5-32B-Instrcut Deepseek-r1-distill Qwen2.5-32B have 58.97% and 86.24% IFR respectively while have pass rate 24.25% and 43.64% on Easy test split.
>    - When we condition on IFR = 100%, reasoning models consistently outperform their chat counterparts.
> 2. **Explanation.** We hypothesize that the lower IFR mainly comes from the difficulty of multi-file generation for reasoning models. In math-style reasoning, the model only needs a single chain of thought to solve one problem, whereas on SWE-Dev it must generate many files and repeatedly switch from one sub-task to another as it starts each new file. These topic shifts make it easy for long reasoning traces to drift away from the original PRD, so reasoning models often omit some required files, leading to a lower IFR. This is consistent with prior findings that chain-of-thought can cause reasoning drift and reduce structural completeness in code [1,2].
>
>
> ---
>
> **[Weakness 2]**
>
> Although RL reportedly improves Pass@1 but reduces diversity, quantitative measures such as output entropy, candidate uniqueness, or AST variance are absent.
>
>
> **Answer**:
>
> We thank the reviewer for highlighting the need for quantitative evidence regarding diversity changes under RL. We provide both qualitative trends and quantitative measurements below.
>
> 1. **Entropy Curves during training process:**
>   We added Figure 19 in the revised submission, demonstrating the policy’s token-level entropy decreases monotonically during RL training. This reflects a contraction in output diversity. We do observe this phenomenon directly during training: the model’s entropy consistently decreases over RL steps, indicating reduced exploration and a sharper output distribution. These trends  fully consistent with the RLVR findings.[3]
> 2. **Quantitative diversity analysis during inference:**
>     - We also sampled 100 tasks from the SWE-Dev and compared a PPO-fine-tuned Qwen2.5-7b-instruct (refered in paper) to the original (non-RL) model. For each task, we drew 16 candidate solutions under identical settings (same sampling temperature, max tokens, etc.). Metrics reported per task were Pass@1 and Pass@5 (success rates), average per-token perplexity, and the unique-pass ratio (the fraction of distinct passing candidates among the 16 samples).
>     - As the table shows, PPO increases success rate (Pass@1 from 10.48% → 11.75%, Pass@5 from 19.00% → 21.20%), while reducing measures of output diversity: average per-token PPL decreases slightly (1.0056 → 1.0034) and the unique-pass ratio falls (15% → 13%).
>
> | Model | **Pass@1** | **Pass@5** | **Avg. per-token PPL** | **Unique-pass ratio** |
> |---|---|---|---|---|
> | **Base** | 10.48% | 19.00% | 1.0056 | 15.00% |
> | **PPO** | 11.75% | 21.20% | 1.0033 | 13.00% |
>
>
> ---
>
> **[Weakness 3]**
>
> The paper defines task complexity via call-tree depth and node count but does not publish explicit thresholds or masking ratios, hindering full reproducibility.
>
>
> **Answer**:
>
> We thank the reviewer for raising this point. To clarify, task complexity metrics are used only for analysis, while dataset construction follows fixed masking rules that do not depend on any thresholds.
> 1. **Masking is independent of complexity thresholds.**
>   For every repository we uniformly apply the same three masking modes: masking depth-1 targets, masking depth-2 dependencies, or masking full call chains for depth ≥ 2.
> 2. **Complexity metrics are only used for analysis.**
>   These rules are fixed and do not involve any depth/node-based filtering, so dataset construction is fully reproducible. The call-tree depth and node count used in Figure 8(a) serve only as analytical difficulty measures for grouping tasks and studying performance trends; they are never used to select or generate tasks.
>
> To avoid ambiguity and facilitate reproducibility, we now explicitly state these rules and the exact bucketing scheme used in our analysis in the Appendix F.3.

---

> ### Author Response · Authors · 2025-11-26
> **[Part 2/3] Author Response**
>
> **[Weakness 4]**
>
> While MAS systems are benchmarked, there is little discussion of communication structure, coordination overhead, or emergent behaviors that might explain observed differences.
>
>
> **Answer**:
>
> We thank the reviewer for highlighting the need for a deeper discussion on communication structure and overhead. We conduct statisital analysis in the table below, which compares seven MAS (all driven by GPT-4o-mini) along four axes: Pass@1, average number of LLM calls, number of roles, and number of agent-to-agent messages.
>
> | MAS System | **Pass@1** | **Avg. LLM Calls** | **#Roles** | **#Avg Agent2agent Com** |
> |---|---|---|---|---|
> | **Reflexion** | 39.77 | 2.12 | 3 | 10.77 |
> | **Self-Refine** | 40.02 | 5.00 | 2 | 26.79 |
> | **Self-Consistentcy** | 37.62 | 6 | 2 | 33.01 |
> | **LLM Debate** | 38.48 | 7 | 2 | 76.35 |
> | **MAD** | 31.50 | 7 | 4 | 2.36 |
> | **AgentVerse** | 38.67 | 4.52 | 4 | 8.83 |
> | **MetaGPT** | 29.56 | 9.69 | 6 | 24.26 |
> | **MapCoder** | 24.55 | 21.01 | 5 | 63.03 |
>
> Two consistent trends emerge:
> 1. **Simple, low-overhead MAS achieve the best trade-off.**
>     General-purpose methods such as Reflexion and Self-Refine use only 2–3 roles and relatively few calls (e.g., Reflexion: 39.77% Pass@1 with 2.12 calls and 10.77 agent-to-agent messages), yet reach the highest accuracy. These systems rely on an iterative self-refinement loop rather than complex division of labor, which keeps the global context concentrated in a small number of agents and reduces coordination cost.
> 3. **Heavy, workflow-driven MAS suffer from coordination overhead and context fragmentation.**
>     Code-specific systems such as MetaGPT and MapCoder employ 5–6 roles, 9.69–21.01 calls, and 24–63 inter-agent messages, but obtain the lowest Pass@1 (29.56% and 24.55%). Every additional handoff between agents introduces opportunities for error propagation and partial loss of context, which is particularly harmful on SWE-Dev tasks that already involve long code contexts and multi-file dependencies.
>
> Taken together, these observations suggest that, on repository-level feature development, **maintaining a stable, unified reasoning context is more important than fine-grained role specialization**. Simple self-refinement style MAS exhibit an emergent behavior of being more robust and sample-efficient, whereas heavily engineered multi-role workflows tend to accumulate overhead without corresponding performance gains.
>
> ---
> **[Question 1]**
>
> How sensitive are results to inference hyperparameters such as temperature, top-p/k, or reflection rounds?
>
> **Answer**:
>
>
> We thank the reviewer for raising this point and respond this in two points:
> 1. **Sensitivity to sampling hyperparameters.**
>     - The table below shows the performance of GPT-4o on SWE-Dev with various temperature and top-p settings.
>     - **Analysis:** Across all tested configurations, the Pass@1 on **both Easy and Hard splits varies only mildly** (typically within 3–4% absolute). For instance, changing the Temperature from 0.1 to 1.0 only resulted in a small performance drop (e.g., Hard Pass@1 dropped from 15.70% to 14.36%).
>
> | **Temperature** | **Top-p** | **Easy Pass@1** | **Hard Pass@1** | **#Agent2agent Communication** |
> |---|---|---|---|---|
> | **0.1** | 0.95 | 43.94% | 15.70% | 10.77 |
> | **0.3** | 0.95 | 41.94% | 14.92% | 26.79 |
> | **0.7** | 0.95 | 39.48% | 14.55% | 33.01 |
> | **1.0** | 0.95 | 40.35% | 14.36% | 76.35 |
> | **0.7** | 0.7 | 41.73% | 15.04% | 2.36 |
> | **0.7** | 0.9 | 43.05% | 15.55% | 8.83 |
>
> A Note on Top-k: We note that due to the current API specifications for the GPT-4o model, we were unable to adjust and evaluate the effect of the top-k setting.
>
> 2. **Sensitivity to reflection rounds.**
>
>     - The table below we shows how performance and average output token length change over rounds on Self-Refine driven by GPT-4o-mini.
>     - **Analysis:** As the number of reflection rounds increases, Pass@1 increases gradually (e.g., Easy: 34.16% $\to$ 39.77%; Hard: 11.76% $\to$ 13.32%). This validates that the iterative refinement process effectively corrects errors and improves solution quality.
>
> | **Rounds R** | **Easy Pass@1** | **Hard Pass@1** | **Avg. Output Length** |
> |---|---|---|---|
> | **0 (no reflection)** | 34.16 | 11.76 | 7840 |
> | **1** | 36.79 | 11.8 | 9217 |
> | **2** | 38.33 | 12.01 | 9808 |
> | **3** | 39.77 | 13.32 | 12020 |
>
> ---
> **[Question 2]**
>
> Can the authors quantify the diversity trade-off in RL models using metrics like output entropy or unique-pass ratios?
>
> **Answer**:
>
> Please refer to Weakness 2, since this point has the same question as Weakness 2.

---

> ### Author Response · Authors · 2025-11-26
> **[Part 3/3] Author Response**
>
> **[Question 4]**
>
> What evidence supports the claim that general multi-agent systems outperform code-specific ones,is this due to communication efficiency or broader role specialization?
>
> **Answer**:
>
> We thank the reviewer for this insightful question and we explain this in three aspects:
> 1. **Excessive Role Specialization:** Systems like MetaGPT, MapCoder, and AgentVerse utilize 5+ specialized roles and rigid, multi-stage message-passing pipelines.
> 2. **Unsustainable Communication Overhead:** This complex coordination causes high communication overhead. As evidenced in Table 3 and Figure 13, their workflows require dozens of inter-agent calls, resulting in the calling time growing dramatically with task complexity (e.g., from 2.12 for Reflexion to 26.61 for ChatDev)—a growth rate of nearly 20 times.
> 3. **Overfitting to Short Tasks:** Because these systems were primarily tuned on small scale coding tasks, like developing a small game within a single file, their complex coordination patterns overfit short tasks and consequently break down on the long, multi-file feature-development problems presented by SWE-Dev.
>
>
> [1] Fu, Tingchen, et al. "Scaling reasoning, losing control: Evaluating instruction following in large reasoning models." arXiv preprint arXiv:2505.14810 (2025).
>
> [2] Zhu, Yuqi, et al. "Uncertainty-Guided Chain-of-Thought for Code Generation with LLMs." CoRR (2025).
>
> [3] Yue, Yang, et al. "Does reinforcement learning really incentivize reasoning capacity in llms beyond the base model?." arXiv preprint arXiv:2504.13837 (2025).
>
> ---
> **Finally:** We really appreciate your insightful suggestions, which have already helped us strengthen both the analysis and presentation of this work. We hope the above clarifications resolve the concerns. If the response is helpful, we would be grateful if you could consider adjusting your score accordingly.

---

### Official Review · Reviewer_59nz · 2025-11-02

**Soundness:** 2
**Presentation:** 2
**Contribution:** 2
**Rating:** 2
**Confidence:** 2

**Summary:**

This paper introduces SWE-Dev, a large-scale dataset aimed at evaluating and training autonomous code-generation systems specifically for feature-driven development within large, real-world repos. The authors provide extensive empirical evaluations across multiple coding systems, chatbot LLMs, reasoning models, and multi-agent systems, demonstrating the challenges of FDD and the utility of SWE-Dev for advancing research in this area.

**Strengths:**

1. This paper studies feature-driven SWE, which is a novel idea.
2. Extensive empirical study, ranging from benchmarking to model training, from SFT to RL..
3. SWE-Dev is a huge dataset (14k samples, all with test cases).

**Weaknesses:**

Major concerns:
1. The experimental section lacks necessary experimental details. For example, the PPO experiments lack a detailed account of important parameters such as reward shaping, max response length, the agent framework employed, and computational resource overhead. They also omit key data from the training process—such as the convergence of training set scores and entropy values. Consequently, the experimental results exhibit limited credibility and reproducibility. It is recommended that parameters of marginal relevance, such as checkpoint saving frequency, need not be elaborated in detail.
2. SWE-Dev is really a big dataset (14k samples), but it also raises concerns about the quality of the data. I believe this paper lacks a direct verification of data quality, such as whether task generation is reliable, whether the tasks are solvable, and whether the test cases and environments are correct.
3. Can training on Swe-Dev generalize to performance improvements on Swe-bench-verified? I believe this is also a good perspective to evaluate the data quality of Swe-Dev.

Minor concerns:
1. The authors seem to have slightly modified the ICLR template, reducing the section spacing on the first four pages.
2. The citation formats in the paper are almost all incorrect.
3. Could you provide more demo cases in SWE-Dev?

**Questions:**

Please refer to "weaknesses".

---

> ### Author Response · Authors · 2025-11-26
> **[Part 1/2] Author Response**
>
> We sincerely appreciate your time and thoughtful comments. Below, we respond to each point in detail.
>
> ---
>
> **[Weakness 1]**
>
> The experimental section lacks necessary experimental details. For example, the PPO experiments lack a detailed account of important parameters such as reward shaping, max response length, the agent framework employed, and computational resource overhead. They also omit key data from the training process—such as the convergence of training set scores and entropy values. Consequently, the experimental results exhibit limited credibility and reproducibility. It is recommended that parameters of marginal relevance, such as checkpoint saving frequency, need not be elaborated in detail.
>
>
> **Answer**:
>
> We thank the reviewer for pointing out these missing details. In the revised version, we add Appendix H.4 to document the PPO setup more thoroughly. The key configurations are:
> - **Reward shaping**: For each rollout solution, the reward is defined as the pass rate: the number of test cases successfully passed by the solution divided by the maximum number of test cases that can be passed.
> - **Compute resources and data scale:** 1 node with 8×A100 GPUs
> - **Sequence lengths:** max prompt length = 8192 tokens; max response length = 2048 tokens.
> - **Key hyperparameters:** actor learning rate = 1e-6, critic learning rate = 1e-5, KL coefficient = 0.001，trained for 5 epochs with batch size 256.
> - **Rollout mechanism:** The RL training does not use any agent framework. Instead, the LLM directly receives the relevant code context and the problem requirement description (PRD) as input and generates a rollout solution.
> - **RL framework:** VERL PPO with a VLLM-based rollout backend.
>
> ---
>
> **[Weakness 2]**
>
> SWE-Dev is really a big dataset (14k samples), but it also raises concerns about the quality of the data. I believe this paper lacks a direct verification of data quality, such as whether task generation is reliable, whether the tasks are solvable, and whether the test cases and environments are correct.
>
>
> **Answer**:
>
> We thank the reviewer for raising this important concern. SWE-Dev is large in scale, but its construction is designed to guarantee that tasks are real, solvable, and evaluated in correct environments.
>
> 1. **Task Generation Quality**
>     - **Real-feature extraction via dynamic tracing.** We use Python execution tracing to obtain the exact call tree exercised by the upstream test cases, ensuring that each task corresponds to a real feature that the project itself validates through its own tests.
>     - **Masking only the ground-truth implementation.** We replace only the traced target implementation with a placeholder while keeping the rest of the project unchanged, guaranteeing that the constructed task faithfully reflects the original feature behavior.
>
>     This ensures that task generation is reliable and grounded in real, test-backed project semantics.
>
>
> 2. **Task Solvability**
>   For every instance, we explicitly verify solvability through a strict fail–pass validation pipeline:
>     - tests pass in the original repository,
>     - tests fail after masking the target implementation,
>     - tests pass again when restoring the ground-truth code.
>
>     This deterministically guarantees that each task is solvable and that the test suite defines a precise behavioral target.
>
> 3. **Test Case & Environment Correctness**
>   These correctness is guaranteed by construction process:
>     - **Test Case Correctness.**
>         - **The test cases are all human-written files from most popular pypi githubs**, which are actively maintained whose frequent updates rely on test suites to validate core functionality. These tests are therefore a realistic and meaningful proxy for correctness in real-world software development.
>         - To assess their discriminative power, we run a test-weakening ablation on 100 sampled task shown in the table below. **We find that removing test cases leads to higher Pass@3, indicating that weaker suites allow more spurious solutions to pass.** This confirms that the original test suites impose meaningful behavioral constraints rather than superficial coverage.
>
>     - **Environment correctness.**
>     We include only repositories whose complete test suites successfully run in our standardized Docker environment.  This ensures all dependencies, imports, and runtime behaviors remain correct and consistent with the original project.
>
> | Avg Testcase per sample | **all** | **6** | **3** | **1** |
> |----------------------------|---------|-------|-------|-------|
> | **pass@1**                 | 52.5%   | 53.2% | 56.5% | 59.0% |

---

> ### Author Response · Authors · 2025-11-26
> **[Part 2/2] Author Response**
>
> **[Weakness 3]**
>
> Can training on Swe-Dev generalize to performance improvements on Swe-bench-verified? I believe this is also a good perspective to evaluate the data quality of Swe-Dev.
>
>
> **Answer**:
>
> We thank the reviewer for this suggestion and conducted experiment to show the generalization of SWE-Dev.
>
> **Experimental setup:** Using an AgentLess setup with gpt-oss-120b (reasoning_effort = high, max output tokens = 32,768), we first collect trajectories on the SWE-Dev training set via rejection sampling and use them to conduct 3 epochs of SFT on Qwen3-8B-Base. For evaluation on SWE-Bench-Verified, we again use the same agentless configuration with a 32,768-token output limit.
>
> **Results:** The zero-shot performance of Qwen3-8B-Base on SWE-Bench-Verified is 1.07%, which increases to 15.73% after SWE-Dev SFT. This even exceeds the off-the-shelf Qwen3-8B (13.80%).
>
> **Conclusion:** Training on SWE-Dev yields substantial and direct improvements on SWE-Bench-Verified, demonstrating that SWE-Dev provides high-quality, transferable supervision useful for downstream code-reasoning and bug-fixing tasks.
>
> |                 model                 | **SWEBench-verifierd** |
> |:-------------------------------------:|:----------------------:|
> | **qwen3-8b**                          |         13.80%         |
> | **qwen3-8b-base**                     |          1.07%         |
> | **qwen3_8b_base_swedev_oss-120b-32k** |         15.73%         |
>
> ---
> **[Weakness 4]**
> Minor concerns:
> - The authors seem to have slightly modified the ICLR template, reducing the section spacing on the first four pages.
> - The citation formats in the paper are almost all incorrect.
> - Could you provide more demo cases in SWE-Dev?
>
> **Answer**:
>
> We thank the reviewer for their careful review and for pointing out these minor concerns regarding the presentation and supporting materials. We appreciate the attention to detail.
>
> **1&2 Template spacing & Citation format**
>
> We will correct all citation and formatting issues in the camera-ready version.
>
> **3 More demo cases**
>
> We have included additional SWE-Dev demo cases in our anonymous project repository(`End-to-end-example.md`), showing the task input, model solution, and execution traces of passing tests.
>
> ---
> **Finally:** We really appreciate your insightful suggestions, which have already helped us strengthen both the analysis and presentation of this work. We hope the above clarifications resolve the concerns. If the response is helpful, we would be grateful if you could consider adjusting your score accordingly.

---

### Official Review · Reviewer_uYqm · 2025-11-04

**Soundness:** 3
**Presentation:** 2
**Contribution:** 3
**Rating:** 4
**Confidence:** 4

**Summary:**

Authors present a new benchmark targeting the training and evaluation of Code LLMs on repo-level feature development tasks. Their dataset provides a execution environment enabling training via RL algorithms. A held-out test set is proposed to evaluate the feature development capabilities of LLMs when applied with different variations of inference algorithms (single/multi agent etc).

**Strengths:**

- Strong motivation: there is a clear need to evaluate complex, repository-level feature development beyond plain code completion/generation.
  - Task complexity is reasonable (≈190 LOC across ~3 files).

- Insights are useful—e.g., MAS appears on par with, or only slightly superior to, simpler systems.

- I find this to be a valuable dataset for open-source/open-weight models to fine-tune on, though the value for evaluation is practically limited.

- Provides a very exhaustive analysis of the dataset and training aspects—a rich set of experiments, with nice insights on single- vs multi-agent systems (inference-based).

- The processed dataset has additional research value:
  - test-quality analysis of these repos (coverage, etc.),
  - proposing improvements to test cases of these popular repos,
  - evaluating LLM-driven optimizations of source code while ensuring functional correctness w.r.t. these tests.

**Weaknesses:**

- Potential test-set contamination: if external models have been trained on these source repos, we can’t guarantee the absence of leakage. I don't think this testing of whether a model can build a feature if it has already seen the source code during pre-training is a reliable signal of the model's performance.

- Test quality is under-analyzed: how confident are we that existing tests confirm correctness? In practice, developers often target coverage rather than correctness. Even a qualitative analysis (e.g., by frontier models) of test thoroughness would help.

- PRD quality/leakage risk: do the PRDs give too many hints to the model for feature development? This warrants a qualitative analysis.

- Train–test split by repository: are splits done at the repo level? Training on method_A from repo_A can help generate method_B from the same repo_A at test time. A more sanitized evaluation would keep train and test repos distinct; otherwise, the evaluation is not fair.

**Questions:**

- Static analysis: why not use static analysis to identify which source functions are called by which tests (and mask the entire call stack)?

- Multiple entrypoints (hypothetical): suppose method_B is called by method_A and tested by test_A, and there is also a test_B calling method_B directly. If you mask m_A and m_B and evaluate only with test_A, the LLM might implement m_A and m_B to pass test_A but still fail test_B. Do you run all implicated entrypoint tests, or just one?

- Line 255: what exactly is the “relevant code context”? How is it selected?

- Spec leakage: isn’t augmenting a docstring with an implementation approach a form of leakage?

- RL details: how are preference pairs constructed for DPO? Why not GRPO as a simpler alternative to PPO?

- Presentation:
  - whitespace between Fig. 1–Fig. 2 and text is too little,
  - 269 — “falls short”,
  - 293 — “development.”

- Role of tests (under-discussed):
  - what about using execution feedback from failing tests to allow the model to re-attempt the feature development task?
  - what about scenarios where tests are unavailable but the feature must be written—can the LLM write tests first and then implement the feature?

---

> ### Author Response · Authors · 2025-11-26
> **[Part 1/3] Author Response**
>
> We sincerely appreciate your time and thoughtful comments. Below, we respond to each point in detail.
>
> ---
>
> **[Weakness 1]**
>
> Potential test-set contamination: if external models have been trained on these source repos, we can’t guarantee the absence of leakage. I don't think this testing of whether a model can build a feature if it has already seen the source code during pre-training is a reliable signal of the model's performance.
>
> **Answer**:
>
> We thank the reviewer for raising the important concern of potential test-set contamination.
>
> 1. **Contamination risk is inherent in public-repo benchmarks, and we do not claim immunity.**
> We fully acknowledge that benchmarks built from public open-source repositories—such as GitHub or PyPI—inevitably face this risk. This limitation is shared by widely used datasets, including SWE-Bench and many prior repository-level coding benchmarks.
>
> 2. **Empirical Evidence Against Widespread Leakage.**
>   Despite the risk, our experimental results suggest that contamination is not the primary factor driving performance:
>     - **Low Performance:** Even the strongest models achieve only ~20% Pass@3 on the hard split. This demonstrates that the tasks remain highly challenging, indicating that contamination is not providing an easy path to a solution.
>     - **Task Complexity:** Even if a model had memorized isolated code fragments from those repositories, they would be largely unusable. The tasks in SWE-Dev require multi-file integration, long-contextual reasoning for feature development, which cannot be solved by simply recalling isolated snippets.
>     - **Synthesized Inputs:** Furthermore, the inputs provided to the model (e.g., the PRDs, code context) are constructed or synthesized through an automated pipeline, rather than being direct copies of existing code, which further mitigates the risk of direct memorization-based cheating.
>
> 3. **Our Automated Method Provides the Systemic Solution for test contamination.**
>     - Our method's core contribution is validating the effectiveness and scalability of this automated methodology to quickly and scalably construct test sets on newly available, model-unseen codebases. We believe it is the necessary prerequisite for achieving this cleaner, more dynamic evaluation paradigm in the future.
>     - This automation allows us to continuously build and update the benchmark using model-unseen, up-to-the-minute repositories, which allows us to systematically evade contamination in the future.
>
> ---
>
> **[Weakness 2]**
>
> Test quality is under-analyzed: how confident are we that existing tests confirm correctness? In practice, developers often target coverage rather than correctness. Even a qualitative analysis (e.g., by frontier models) of test thoroughness would help.
>
>
> **Answer**:
>
> We appreciate the concern regarding test quality variability. We will respond in the following three aspects:
> 1. **Repository selection.** SWE-Dev uses popular, actively maintained PyPI packages whose test suites are continuously relied upon to validate real-world functionality. Consequently, these established projects heavily rely on comprehensive test suites to validate their core functionality, making these tests a realistic and meaningful proxy for correctness in real-world software development.
> 2. **Test-weakening ablation.** To assess their discriminative power, we run a test-weakening ablation on 100 sampled tasks and observe that removing test cases consistently increases Pass@3. This shows that the original test suites are indeed enforcing stricter behavioral correctness and are not merely superficial coverage tests.
> 3. **Dynamic coverage analysis.** Moreover, our dynamic tracing confirms that these tests exercise substantial portions of the feature’s call tree, rather than only shallow paths. On the test set, they touch 5.82 functions across 2.98 files on average (6.03 functions / 2.51 files for train), as shown in Figure 10 and Table 2, demonstrating broad behavioral coverage.
>
> | Number of testcase per sample | **1** | **3** | **6** | **all** |
> |-------------------------------|-------|-------|-------|---------|
> | **pass@1**                    | 59.0% | 56.5% | 53.2% | 52.5%   |

---

> ### Author Response · Authors · 2025-11-26
> **[Part 2/3] Author Response**
>
> **[Weakness 3]**
>
> PRD quality/leakage risk: do the PRDs give too many hints to the model for feature development? This warrants a qualitative analysis.
>
>
> **Answer**:
>
> We thank the reviewer for raising the crucial question. We have conducted qualitative and quantitative analyses to directly address both aspects of this concern:
>
> 1. **PRD quality.**
>   We evaluate PRD quality through a human study on 100 sampled tasks in Section 5, where two experienced engineers compared the original and refined PRDs across Actionability, Completeness, and Clarity. As shown in Figure 8c, the refined PRDs consistently score higher on all three dimensions. This confirms that our refinement process yields clearer and more usable requirements without reducing the task difficulty.
> 2. **PRD Leakage Risk**
>     - In constructing the PRDs, we explicitly ensure that they contain only behavioral information already present in the repository (from docstrings, tests, and surrounding context) and no implementation details.
>     - Using repository documentation elements, such as docstrings, to specify functionality without revealing the internal logic is a common and accepted practice in code generation benchmarks, including HumanEval and MBPP. Our approach follows this principle at the repository level.
>
>
> ---
> **[Weakness 4]**
>
> Train–test split by repository: are splits done at the repo level? Training on method_A from repo_A can help generate method_B from the same repo_A at test time. A more sanitized evaluation would keep train and test repos distinct; otherwise, the evaluation is not fair.
>
>
> **Answer**:
>
> Yes, SWE-Dev strictly split train and tests set at the repository level: all repositories used for training are disjoint from those in the test set. This design explicitly avoids the data leakage from train set and ensures that our evaluation measures generalization to unseen repositories. We will highlight this repo-level split more prominently in the revised version to avoid misunderstanding.
>
> ---
> **[Question 1]**
>
> Static analysis: why not use static analysis to identify which source functions are called by which tests (and mask the entire call stack)?
>
>
> **Answer**:
>
> We did not adopt a purely static AST-based approach because, in real-world Python projects, static AST analysis can only provide an approximate call graph and **cannot reliably recover the exact set of functions that are actually executed by a given test**. Specifically:
>
> 1. **Static AST analysis cannot reliably capture Python’s dynamic behavior.**
>
>     In real projects, many call targets are only determined at runtime, e.g.,
>     - **Dynamic imports**, such as `importlib.import_module(configured_name)` or `__import__(name)`, where the module name comes from configuration.
>    - **Plugin registration**, where string names from `entry_points` or config files are mapped to concrete functions/classes and loaded at runtime.
>    - **Reflection-based dispatch**, e.g., `getattr(module, func_name)` or `globals()[name]` deciding which function to call.
>
>     These behaviors depend on runtime configuration, environment, and data that are not fully encoded in the local AST.
>
> 2. **Static AST has difficulty reconstructing framework/workflow-driven control flow.**
>
>     In many Python projects, control flow from tests to application code is mediated by frameworks rather than explicit function calls written by the user, for example:
>    - **`pytest`** automatically discovers `test_*.py` functions, runs them through its internal runner, and injects fixtures and hooks before invoking user code.
>    - **Web/CLI frameworks** use decorators or registration APIs (e.g., `@app.route("/user")`, `@cli.command()`) to register handlers; at runtime, the framework routes a request or command to the appropriate handler and calls it.
>
> ---
> **[Question 2]**
>
> Multiple entrypoints (hypothetical): suppose method_B is called by method_A and tested by test_A, and there is also a test_B calling method_B directly. If you mask m_A and m_B and evaluate only with test_A, the LLM might implement m_A and m_B to pass test_A but still fail test_B. Do you run all implicated entrypoint tests, or just one?
>
>
> **Answer**:
>
> We thank the reviewer for the question. Each SWE-Dev task is designed to evaluate one specific feature, and all tests that verify this feature are included in that task.
>
> If multiple test cases (e.g., test_A and test_B) pertain to the same feature behavior, they are grouped into the same task and must all be satisfied.
>
> Conversely, if test_B exercises a different feature, then it is intentionally outside the scope of this task—similar to how SWE-Bench focuses on resolving the tests associated with the target issue rather than validating unrelated functionality.

---

> ### Author Response · Authors · 2025-11-26
> **[Part 3/3] Author Response**
>
> **[Question 3]**
>
> Line 255: what exactly is the “relevant code context”? How is it selected?
>
> **Answer**:
>
> The "relevant code context" here refers the minimal set of source files the model must read to complete the task. Since task's call tree enumerates the functions invoked by its tests, and each function node encodes its source file. We simply take the set of file names appearing in the task's call tree as the smallest file subset needed for implementing the feature.
>
> ---
> **[Question 4]**
>
> Spec leakage: isn’t augmenting a docstring with an implementation approach a form of leakage?
>
> **Answer**:
>
> Please refer to Weakness 3.
>
> ---
> **[Question 5]**
>
> RL details: how are preference pairs constructed for DPO? Why not GRPO as a simpler alternative to PPO?
>
> **Answer**:
>
> 1. **DPO data**
>     In DPO, the preferred sample is the ground-truth implementation, and the rejected sample is the model-generated candidate that fails on tests.
>
> 2. **Why not GRPO**
>
> We chose PPO primarily to validate that SWE-Dev provides stable, execution-based reward signals for RL. PPO is a well-established baseline in code RL, and suffices to demonstrate that our dataset supports accurate test-driven feedback; exploring alternatives such as GRPO is left for future work.
>
> ---
> **[Question 6]**
>
> Presentation:
>
> - whitespace between Fig. 1–Fig. 2 and text is too little,
> - 269 — “falls short”,
> - 293 — “development.”
>
> **Answer**:
>
> We thank the reviewer for detailed checking, we have corrected this in revised submission.
>
> ---
> **[Question 7]**
>
> Role of tests (under-discussed):
>
> - what about using execution feedback from failing tests to allow the model to re-attempt the feature development task?
> - what about scenarios where tests are unavailable but the feature must be written—can the LLM write tests first and then implement the feature?
>
> **Answer**:
>
> We appreciate the reviewer’s suggestions.
>
> 1. **Using execution feedback for re-attempts.**
> We conducted experiment with this setting through a self-refine procedure shown in the table below. With GPT-4o-mini, reflection rounds do improve performance (e.g., Easy Pass@1: 34.16→39.77 and Hard: 11.76→13.32), but gains remain modest given current model capability.
>
> 2. **Writing tests when none are available.**
> This is an interesting direction but outside SWE-Dev’s current scope, which focuses on feature implementation given developer-authored tests. Extending the benchmark to include test-generation + implementation workflows is a promising future line of work.
>
> | Rounds R              | **Easy Pass@1** | **Hard Pass@1** |
> |-----------------------|-----------------|-----------------|
> | **0 (no reflection)** | 34.16           | 11.76           |
> | **1**                 | 36.79           | 11.8            |
> | **2**                 | 38.33           | 12.01           |
> | **3**                 | 39.77           | 13.32           |
>
> ---
> **Finally:** We really appreciate your insightful suggestions, which have already helped us strengthen both the analysis and presentation of this work. We hope the above clarifications resolve the concerns. If the response is helpful, we would be grateful if you could consider adjusting your score accordingly.

---

### Author Response · Authors · 2025-12-03
**Rebuttal Summarization**

We sincerely thank all reviewers and the committee for their time. We summarize SWE-Dev’s core contributions and how our updates address reviewer concerns.

**Contribution**

SWE-Dev introduces a large-scale benchmark for autonomous end-to-end feature-driven software development. Each instance consists of a natural-language project requirement document (PRD), a partially masked real-world Python repository, and developer-authored unit tests in a runnable Docker environment, enabling execution-based evaluation and training at repository scale (14k train, 500 test instances over 1,061 packages, with ≈190 LOC modified across ≈3 files per task).

We evaluate our dataset on single LLM and multi-agent setting and checking the dataset feasibiltiy via SFT, RL and multiagent finetuning. Experiments shows that feature development is still a hard task for current LLM and training even on small model helps on this task.

**Comparison with related work** (Reviewer icEc W2–W4)
- **Task differences.** We added a comparison table against related works. Existing datasets focus on bug fixing, issue solving, or performance tuning, usually involving small, localized edits. In contrast, SWE-Dev targets feature implementation from PRDs, with **larger, multi-file code changes** that **add new features**.
- **Construction methodology differences.** We clarify that prior datasets often rely on static AST transformations, commit-based mining, or manually curated performance harnesses. SWE-Dev instead uses dynamic execution tracing on developer-written tests to obtain the exact call tree, masks only the traced implementation. This yields feature-level, executable tasks at scale with real tests and repositories, complementary to prior SWE benchmarks.

**Data quality**
1. **PRD quality** (Reviewer uYqm W3). PRDs are constructed from existing documentation (docstrings, comments, test descriptions) and then refined. A small human study (Fig. 8c) shows that PRD's quality in actionability, completeness.
2. **PRD leakage** (Reviewer uYqm W3, Q4). We emphasize that PRDs describe only externally observable behavior already documented in the repository, not internal code. Using docstrings and documentation as behavioral specifications is standard in code benchmarks like HumanEval and MBPP; SWE-Dev follows this practice at the repository level, so PRDs provide realistic guidance rather than leaking ground-truth implementations.
3. **Contamination risk and splits** (Reviewers uYqm W1, W4; 59nz W2). As with other public-repo benchmarks, we do not claim zero pretraining contamination. We mitigate this by strictly splitting train and test at the repository level and by constructing tasks via masking and synthesized PRDs rather than copying GitHub issues verbatim. Empirically, even frontier models achieve only modest Pass@k on hard, multi-file tasks, which suggests that direct memorization is not driving performance.
4. **Language-agnostic methodology** (Reviewer icEc W7). While the current implementation uses Python tracing, the construction only assumes runtime instrumentation and function-level call tracing, which are available in many ecosystems (e.g., JVM, Go, JS/TS, C/C++).

**Extended experiments**
1. **Generalization to SWE-Bench-Verified** (Reviewer 59nz W3). We perform SFT on Qwen3-8B-Base on SWE-Dev, then evaluate with an Agentless setup on SWE-Bench-Verified. Performance improves from 1.07% to 15.73%, surpassing off-the-shelf Qwen3-8B-Instruct, indicating the generalization ability.
2. **Agentic baselines** (Reviewer icEc W6). We report results for mainstream SWE agents (Agentless and OpenHands) on the SWE-Dev test set.
3. **RL training reporducability** (Reviewers icEc W9; 59nz W1). We added detailed PPO configuration (reward definition based on test pass rate, rollout protocol, sequence lengths, compute budget, and hyperparameters) and report training curves in Appendix H.4 and Fig. 17 and Fig. 18. This addresses concerns about the sufficiency and reproducibility of the RL setup.

**Analysis and insights**
1. **Reasoning models vs chatbots** (Reviewers icEc W8, pjSU W1, uYqm Q2). In Fig. 14, we analyze Instruction Following Rate (IFR)—the share of tasks where all PRD-required files are produced. Reasoning models show lower IFR than chat models, especially on multi-file tasks; yet when IFR = 100%, they outperform chat models. This indicates that their weakness stems from incomplete adherence to the required multi-file format rather than insufficient reasoning ability.
2. **MAS communication structure and efficiency** (Reviewer pjSU W4, Q4). Our statistics on Pass@1, roles, LLM calls, and messages show that simple, low-overhead MAS outperform heavily specialized, communication-heavy systems on SWE-Dev.
3. **Hyperparameter** (Reviewer pjSU Q1; uYqm Q7). Temperature/top-p ablations for GPT-4o produce only small changes in Pass@1, while adding self-reflection rounds steadily improves Pass@1 at the cost of longer outputs.

---

### Note · Program_Chairs · 2026-01-17
**Submission Desk Rejected by Program Chairs**

The following references in this submission do not refer to real documents and/or have major errors in bibliographic information:

     Luan Xavier, Marco D'Ambros, André Hora, Romain Robbes, and Marco Tulio Valente. Characterizing and comparing the distribution of software evolution tasks in industrial and open source projects. In 2017 IEEE 24th International Conference on Software Analysis, Evolution and Reengineering (SANER), pp. 130-140. IEEE, 2017. doi: 10.1109/SANER.2017.7884614.